# Unveiling the contribution of particle-associated non-cyanobacterial diazotrophs to N₂ fixation in the upper mesopelagic North Pacific Gyre
Christian F. Reeder [1,2,3], Alba Filella [1,2,4], Anna Voznyuk [5], Arthur Coët [1,2], Reece C. James[6], Tully Rohrer [6], Angelicque E. White[6], Léo Berline[1], Olivier Grosso[1], Gert van Dijken [7], Kevin R. Arrigo [7], Matthew M. Mills [7], Kendra A. Turk-Kubo [5] ✉ & Mar Benavides [1,2,8] ✉

Dinitrogen (N₂) fixation supports marine life through the supply of reactive nitrogen. Recent studies suggest that particle-associated non-cyanobacterial diazotrophs (NCDs) could contribute significantly to N₂ fixation contrary to the paradigm of diazotrophy as primarily driven by cyanobacterial genera. We examine the community composition of NCDs associated with suspended, slow, and fast-sinking particles in the North Pacific Subtropical Gyre. Suspended and slow-sinking particles showed a higher abundance of cyanobacterial diazotrophs than fast-sinking particles, while fast-sinking particles showed a higher diversity of NCDs including *Marinobacter*, *Oceanobacter* and *Pseudomonas*. Using single-cell mass spectrometry we find that Gammaproteobacteria N₂ fixation rates were higher on suspended and slow-sinking particles (up to 67 ± 48.54 fmol N cell⁻¹ d⁻¹), while putative NCDs' rates were highest on fast-sinking particles (121 ± 22.02 fmol N cell⁻¹ d⁻¹). These rates are comparable to previous diazotrophic cyanobacteria observations, suggesting that particle-associated NCDs may be important contributors to pelagic N₂ fixation.

Dinitrogen (N₂) fixation is a key process for introducing bioavailable nitrogen in the ocean supporting primary productivity[1,2]. This process is orchestrated by microorganisms called "diazotrophs". Traditionally, N₂ fixation was mainly attributed to cyanobacteria residing in the sunlit surface waters of the ocean[3,4]. However, recent research has unveiled an extensive diversity and widespread distribution of non-cyanobacterial diazotrophs (NCDs) in different marine habitats[5–11]. Unlike their cyanobacterial counterparts, NCDs are not confined to sunlit surface waters, but are rather distributed across a wide array of pelagic habitats such as upwelling regions[12], oxygen minimum zones[8,13], the deep ocean[14], and temperate coastal zones[15]. The process of N₂ fixation demands substantial energy, with a common vulnerability shared among all diazotrophs which is the irreversible inactivation of nitrogenases by oxygen[16]. Consequently, diazotrophs allocate significant resources to safeguarding nitrogenases from oxygen exposure. Although cyanobacteria have developed various mechanisms to

evade oxygen inactivation[17,18], knowledge about the strategies employed by NCDs remains limited. One hypothesis, however, suggests that marine snow or other organic particles can create favorable conditions for NCDs by offering a low-oxygen environment as well as carbon and energy resources (e.g., particles can have high labile carbon content and a high C:N stoichiometry)[6,19–25]. Recent studies have shown that particles stimulate pelagic N₂ fixation when resuspended from sediments in neritic ecosystems[26,27] and that NCDs have the genomic potential to seek out for particles through chemotaxis[28], and subsequently colonize[29], and fix N₂ on them[30]. To date, however, the magnitude of particle-associated NCDs N₂ fixation and its contribution to the oceanic nitrogen reserves remains unquantified[31].

Through *nifH* gene sequencing and quantitative PCR, NCDs have been associated with sinking and suspended particles in the North Pacific[9], and in the upwelling regions of the eastern tropical South Pacific Ocean[31]. NCDs

¹Aix Marseille Univ, Université de Toulon, CNRS, IRD, MIO UM 110, Marseille, France. ²Turing Centre for Living Systems, Aix-Marseille University, Marseille, France. ³Faculty of Health and Life Sciences, Department of Biology and Environmental Science, Ctr Ecol & Evolut Microbial Model Syst (EEMiS), Linnæues University, Kalmar, Sweden. ⁴Department of Molecular and Cellular Biology, The University of Arizona, Tucson, AZ, USA. ⁵University of Santa Cruz, Ocean Sciences Department, Santa Cruz, CA, USA. ⁶Department of Oceanography, University of Hawaii Manoa, Honolulu, HI, USA. ⁷Department of Earth System Science, Stanford University, Stanford, CA, USA. ⁸National Oceanography Centre, Southampton, UK. ✉e-mail: kturk@ucsc.edu; mar.benavides@noc.ac.uk

have been widely detected in the 0.2–5 μm, 5–20 μm, 20–180 μm and 180–2000 μm filter size fractions of the TARA Ocean metagenomes dataset, with ~50% of *nifH* reads in the 0.5–5 μm size-fraction and up to 25% of total *nifH* reads in the 180–2000 μm size fractions[32,33]. NCDs' metagenome-assembled genomes covering the size range 0.8–2000 μm have been shown to be more abundant than their cyanobacteria counterparts in the surface ocean[32]. This does add to the growing evidence that NCDs often present a particle-associated lifestyle, albeit knowledge of their $N_2$ fixation potential is limited to direct isotope tracing measurements. Using CARD-FISH combined with single-cell mass spectrometry, we measure specific $N_2$ fixation rates of Gammaproteobacteria NCDs and other "putative NCDs". $N_2$ fixation activity in the upper mesopelagic zone (150 m) of the North Pacific Ocean. By distinguishing different particle types (suspended, slow, and fast sinking), our results provide insights into niche separation of NCDs and the role of Gammaproteobacteria in oceanic nitrogen input.

## Materials and methods
### Hydrography and sampling
The NCD cruise took place in the North Pacific between 4 June and 6 July 2022 onboard the R/V *Kilo Moana* (cruise KM2206; Fig. S1). Water column profiles of temperature, salinity, fluorescence, beam transmission, oxygen, and photosynthetic active radiation down to ~500 m were obtained with a Seabird SBE 9/11plus CTD, PAR sensor, and transmissometer mounted on a 24-Niskin bottle rosette.

Samples for the measurement of nitrate plus nitrite ($NO_3^- + NO_2^-$), phosphate ($PO_4^{3-}$), and silicic acid ($Si(OH)_4$) concentrations were collected at 150 m and filtered through pre-combusted (450 °C for 4.5 h) 25 mm GF/F filters and stored in acid-cleaned bottles at −20 °C until analysis using standard techniques on a Seal Analytical AA3 HR Nutrient Autoanalyzer at the University of Hawaii at Manoa's SOEST Laboratory for Analytical Biogeochemistry. Samples for fluorometric analysis of chlorophyll-*a* (Chl *a*) were measured from the same depths using a Turner Fluorometer TD700 (Turner Designs, Inc.), according to N. Welschmeyer[34] and using spinach chlorophyll standard from Sigma-Aldrich (item C5753).

Suspended (SUSP), slow sinking (SS) and fast sinking (FS) particles were collected using a marine snow catcher (MSC; OSIL, Havant, UK) deployed to 150 m at twelve stations (Fig. S1), as described in Riley et al.[35]. This depth was chosen as a transition between the epipelagic and mesopelagic layers. Upon recovery, the MSC was secured on deck, protected from sunlight, and particles allowed to settle (i.e., particles being separated based on sinking speed) for 4 h, as described in Riley et al.[35], with the modification that particles were settled for 4 h and not 2 h. This was done due the assumption of lower biomass at 150 m depth. After the 4 h period, the SUSP and SS fractions were collected using acid-clean tubing in six replicate polycarbonate bottles of 4500 ml and 500 ml, respectively (Nalgene, Rochester, NY, USA). For DNA analyses, the SUSP and SS fractions were collected using acid-clean tubing in triplicate polycarbonate bottles of 4500 ml and 500 ml, respectively (Nalgene, Rochester, NY, USA). The FS fraction (total ~300 ml) was sampled using sterile serological pipettes and pooled in a 500 ml polycarbonate bottle. From this pooled FS fraction, three 30 ml replicates were used for DNA analyses and three 60 ml replicates were used for single-cell $N_2$ fixation measurements (see below).

### $^{15}N_2$ incubations
Triplicate sub-samples from each MSC fraction (4500 ml, 500 ml, and 60 ml from the SUSP, SS, and FS fractions respectively) were collected in transparent polycarbonate bottles with Teflon-coated septum screw-caps (Nalgene, Waltham, MA, USA) as described above, and spiked with 10% v/v $^{15}N_2$-enriched filtered seawater (Cambridge Isotopes Inc., Tewksbury, MA, USA) according to the dissolution method as described in White et al.[36]. The volumes incubated for each fraction vary as a result of the differences in volume available from the MSC fractions. The bottles were incubated for 24 h in the dark temperature-controlled incubator at temperatures corresponding to 150 m at each station sampled (Fig. S1).

After incubation, 500 ml, 50 ml, and 10 ml were subsampled from the SUSP, SS, and FS fractions, respectively, for Catalyzed Reporter Deposition Fluorescence In Situ Hybridization (CARD-FISH) and nanoscale Secondary-Ion Mass Spectrometry analyses (nanoSIMS) (see below). Incubated seawater aliquots of each replicate and MSC fraction were transferred to Exetainer tubes for membrane inlet mass spectrometry analyses to determine the $^{15}N$ at% enrichment of the $N_2$ in the incubation seawater[37]. The remaining volume of each MSC fraction (4000 ml, 450 ml, and 120 ml) was filtered onto pre-combusted (450 °C, 4 h) glass fiber filters (GF/F, Whatman) and dried for 24 h at 60 °C for particulate carbon and nitrogen analyses (PC and PN, respectively). Filters were stored at room temperature until analysis on an elemental analyzer spectrometer (INTEGRA 2, SerCon Ltd, Crewe, UK). PC and PN concentrations were corrected for the volume of each MSC fraction according to Riley et al.[35]. The analytical precision associated with mass determination ranged between 0.8 and 4.8% of PC and between 0.2 and 2.8% of PN.

### Identification of NCDs using CARD-FISH
Filters for targeted-NanoSIMS (i.e., CARDFISH + NanoSIMS) analyses were conducted on sub-samples of the $^{15}N_2$ incubations described above. Volumes of 500 ml, 50 ml, and 10 ml for the SUSP, SS, and FS fractions, respectively, were filtered onto 0.2 μm polycarbonate filters (Nuclepore, Whatman, Maidstone, UK), fixed with 16% microscopy grade paraformaldehyde (1.6% final concentration) and stored at −80 °C. These filters were used to identify Gammaproteobacteria cells using a CARD-FISH assay (see below). CARD-FISH positively stained cells were then mapped for single-cell isotope ratio measurements using nanoSIMS (see Section "Single-cell N2 fixation rates"). Gammaproteobacteria were chosen as targets due to its prevalence in these waters[38,39].

Filter scissions were embedded in 0.1% ultrapure agarose (Life Technologies, Carlsbad, CA, USA). This was followed by two-step permeabilization using a 10 mg ml$^{-1}$ lysozyme and 60 U ml$^{-1}$ achromopeptidase solution incubated at 37 °C for 1 h and 30 min, respectively. Hybridization was carried out with horseradish peroxidase-labeled oligonucleotide probes (Biomers.net Inc., Ulm/Donau, Germany) targeting Gammaproteobacteria at 46 °C. The probes used to target Gammaproteobacteria were GAM42A, as named in probebase[40]. Following the hybridization at 46 °C the filters were washed with washing buffer (i.e., 47.775 ml Milli-Q + 700 μl 5 M NaCl + 1 ml 1 M TRIS HCl + 0.5 ml 0.5 M EDTA + 25 μl 20% SDS) at 48 °C to remove unincorporated probes. The tyramide signal amplification (TSA) step consisted of Alexa 488 fluorophore (Biomers.net, Ulm, Germany) diluted in amplification buffer (final concentration: 1X PBS, 1 mg ml$^{-1}$ blocking agent, 2 M NaCl, 100 mg ml$^{-1}$ dextran sulfate) and hydrogen peroxide (0.0015% final concentration). After the TSA step, filters were washed with 1X PBS, 0.01 M HCl and rinsed with autoclaved Milli-Q water. Filters were then dried and counter-stained with 4',6-diamidino-2-phenylindole (DAPI) with ProlongTM Diamond Antifade Mountant (Molecular Probes, Eugene, OR, USA). Filter slices were visualized on a Zeiss Axioplan epifluorescence microscope (Oberkochen, Germany) to check for positive hybridized cells on particles. Filters were then gently washed with Milli-Q water and placed upside down on a silicon wafer (1.2 × 1.2 cm, with a 1 × 1 mm raster, Pelotec SFG12 Finder Grid substrate, Ted Pella, Redding, CA, USA), then frozen at −80 °C for ~5 min. Filters were gently removed from the wafers while still frozen, facilitating the transfer of cells and particles to the wafer. Wafers were then stored at −20 °C until further analyses. Before nanoSIMS analysis, the wafers were allowed to dry before mapping target cells using an epifluorescence microscope with 10, 40, and 60X dry objectives, by targeting DAPI (Ex: 350 nm/Em: 465 nm) and Alexa488 (Ex: 488 nm/Em: 591 nm) on a Zeiss Axioplan epifluorescence microscope (Oberkocken, Germany) at UCSC. Finally, the particles containing positively stained cells by the CARD-FISH assay were counted and their size measured using the Zeiss ZEN microscopy software.

## Single-cell N$_2$ fixation rates

Stations used for nanoSIMS analysis (S06, S07, S09, S20, and S24) were selected based on their spatial location (Fig. S1) to cover a wide range of biogeochemical conditions, and based on the relative abundance of NCDs in each MSC fraction (Fig. 3). Particles previously mapped by microscopy were located using the CCD camera on the Cameca nanoSIMS 50 L at the Stanford Nano Shared Facilities (Stanford, CA, USA). Briefly, images were then rastered with a 20 keV cesium primary ion beam (~5 pA), focused into a ~120 nm spot with a mass resolving power of >9000. Images of $^{12}C^-$, $^{13}C^-$, $^{12}C^{14}N^-$, $^{12}C^{15}N^-$, $^{34}S^-$, and $^{12}C_2^-$ were collected over 40–60 planes over an area of 20–50 µm² and a resolution of 254 ×254 pixels with a dwell time of 1 ms per pixel[41]. The image analysis software Look@nanoSIMS[42] was used to process isotope images. Corrections for beam and stage drift were made for all scans before the planes were accumulated and cells on particles were selected as regions of interest (ROIs). One hundred eighteen cells were detected, however only those ROIs with ≥50 ion counts on the $^{12}C^{15}N^-$ channel and a Poisson error <5% were analyzed leading to 67 cells being considered for further data analyses. Single-cell N$_2$ fixation rates (fmol N cell$^{-1}$ d$^{-1}$) were calculated according to Turk-Kubo et al.[43]:

$$\text{Single cell } N_2 \text{ fixation rate calculation} = \frac{A_{final} - A_{start}}{A_{N_2} - A_{start}} \frac{PN_{cell}}{\Delta T}$$

Where $A_{final}$ and $A_{start}$ are the $^{15}N$ atom% of enrichment at time final and zero, respectively. $A_{N2}$ is the $^{15}N$ atom% enrichment of the N$_2$ source pool, $\Delta T$ is incubation time (days) and $PN_{cell}$ is particulate nitrogen content per cell. $A_{start}$ was calculated to be −23 per mil off the natural abundance. $PN_{cell}$ was calculated from the biovolume of single cells and conversion factors. Cell biovolume (BV) was estimated assuming a spherical cell:

$$BV = \frac{\pi}{6} \text{width}^3$$

Carbon content per cell ($PC_{cell}$) was determined according to Verity et al.[44] for cells >0.6 µm³:

$$PC_{cell} = 433BV^{0.863}$$

Nitrogen content per cell ($PN_{cell}$) was determined from $PC_{cell}$ values using a C:N ratio of 5.2, adjusted for heterotrophic cells according to Vrede et al.[45]. Finally, the $PN_{cell}$ for cells below <0.6 µm³ was determined according to Khachikyan et al.[46]:

$$PN_{cell} = 197BV^{0.46}$$

## DNA extractions, *nifH* sequencing, and bioinformatics

Samples for DNA were filtered onto 0.2 µm polysulfone filters (Supor, Pall Gelman, Port Washington, NY, USA), transferred to sterile bead beater tubes containing a mix of 0.1- and 0.5-mm glass beads and stored at −80 °C. DNA was extracted at sea using a protocol modified for rapid DNA extraction (after Preston et al.[47]), and purified DNA was stored at −80 °C. DNA concentration and quality were screened using a NanoDrop (Model One C, Madison, WI, USA). Partial *nifH* fragments were amplified using a universal nested *nifH* PCR assay[48,49], as detailed in Cabello et al.[50]. Second round PCR primers were synthesized with a 5' common sequence linker[51] and used to create barcoded libraries following the targeted amplicons sequencing approach described in Green et al.[52] at the DNA Service Facility at the University of Illinois at Chicago, USA. Amplicons were sequenced bidirectionally (2 × 300 bp) using Illumina MiSeq technology at the W.M. Keck Center for Comparative and Functional Genomics at the University of Illinois at Urbana-Champaign, USA.

Raw reads were processed in R using the DADA2 pipeline v1.29[53]. Briefly, reads were quality checked, filtered, and trimmed. This was followed by dereplicating and merging of the reads and finally removal of chimeras.

Taxonomy was assigned using the *nifH* DADA2 database v2.0.5[54]. Poorly assigned sequences (bootstrap below 80% at order level) were filtered out. Sequences were deposited in NCBI with Bioproject number PRJNA1085235. Diversity indices and sample richness (i.e., Shannon, Simpson, Chao1) were calculated using R packages ampvis2 v2.8[55] and vegan[56]. Correlations between environmental factors (temperature, salinity, oxygen, PC, PN, $PO_4^{3-}$, $Si(OH)_4$), $NO_3^- + NO_2^-$ and Chl $a$ concentrations with particle size-fraction abundance, bulk particle-associated N$_2$ fixation and single-cell N$_2$ fixation rates, were checked for the top 10 most abundant ASVs (Amplicon Sequence Variant) using Pearson correlation with a significance threshold of 0.05[57].

## Statistics and reproducibility

A detailed overview of statistical analyses are given in the respective sections of the "results". In brief, 59 particles were analyzed and 19 particles contained for $^{15}N_2$ enriched cells which added up to 53 Gammaproteobacteria enriched cells and 14 putative NCD cells. A Tukey test was used to test for significant differences between N$_2$ fixation rates and the MSC fractions. Differences among *nifH* gene sequence reads were explored using Pearson correlation with a $p$ value of 0.05.

## Reporting summary

Further information on research design is available in the Nature Portfolio Reporting Summary linked to this article.

# Results

## Environmental conditions

Relatively high Chl $a$ concentrations and lower temperatures were observed at higher latitudes during the cruise (>32°N; Fig. S1), when compared to lower latitudes (<32°N; Fig. S1). Chl $a$ peaks were observed at different depths across stations, including a shallow peak (~0.6 µg l$^{-1}$) at 60–70 m (S06, S07), a peak (~0.3 µg l$^{-1}$) at 100 m (S04, S09, S11, S20, S22, S26), and a deep peak (~0.3 µg l$^{-1}$) at 130 m (S01, S02, S24, S28) (Fig. S1). Temperature decreased with depth from 15–27 °C at the surface to 12–22 °C at 150 m. At 150 m temperatures were >20 °C at stations S01, S02, S26, S27, and S28, while at stations S04, S09, S11, S20 observed temperatures ranged between 15 °C and 20 °C. At station S07 the temperature was <15 °C (Fig. S1). At 150 m, $PO_4^{3-}$ concentrations ranged between 0.27 and 0.48 µmol l$^{-1}$ at stations S06, S07, S09, S20, and S22, together with $NO_3^- + NO_2^-$ concentrations ranging between 2.7 and 6.4 µmol l$^{-1}$. On the contrary, lower $PO_4^{3-}$ and $NO_3^- + NO_2^-$ concentrations were observed at stations S01, S02, S11, S24, and S28 where they ranged between 0.045–0.11 µmol l$^{-1}$ and 0.04–0.64 µmol l$^{-1}$, respectively. $Si(OH)_4$ values ranging between 5 and 11 µmol l$^{-1}$ in S06, S07, and S09, while values were between 1.3 and 3.8 µmol l$^{-1}$ in the remaining stations (Supplementary Data S1).

## Particle-associated N$_2$ fixation rates

Five stations, namely S06, S07, S09, S20, and S24, were selected for nanoSIMS analyses based on their geographic location and relative NCD abundance. Particle sizes as measured by microscopy were 19 µm to 295 µm, 30 µm to 150 µm, and 27 to 80 µm in the SUSP, SS, and FS fractions, respectively (see Section "Identification of NCDs using CARD-FISH", Supplementary Data S2, S3). Cells not hybridized by the CARD-FISH assay (i.e., not Gammaproteobacteria) also showed $^{15}N$ enrichment. These were designated as "putative NCDs" according to Harding et al.[30]. Among the 59 particles analyzed, 19 contained $^{15}N$-enriched cells (including both gammaproteobacterial and putative NCDs) (Fig. 1, Supplementary Data S2, S3). We did not find any particles with N$_2$ fixing cells at stations S07 and S24. Hence, we only refer to stations S06, S09, and S20 from now on. Gammaproteobacterial cells significantly enriched in $^{15}N$ were only observed at stations S06, S09, and S20, with abundances ranging up to $0.4 \pm 0.14$ cells particle$^{-1}$, being highest in the SUSP fraction (Supplementary Data S2), while the abundance of putative NCDs ranged between zero and $0.07 \pm 0.04$ cells particle$^{-1}$ (Supplementary Data S3).

**Fig. 1 | CARD-FISH and nanoSIMS analyses of particle-associated cells.** Example of catalyzed reporter deposition fluorescence in situ hybridization (CARD-FISH) (**A**) and nanoSIMS (**B**, **C**) from Station 20. nanoSIMS (**B**) shows the ratio of $^{12}C^{15}N$-:$^{12}C^{14}N$- indicating cells enriched with $^{15}N$ and **C** shows $^{12}C$, $^{13}C$, $^{12}C_2$, $^{12}C^{13}C$, $^{12}C^{14}N$, $^{12}C^{15}N$, $^{13}C$ and secondary electrons (Esi).

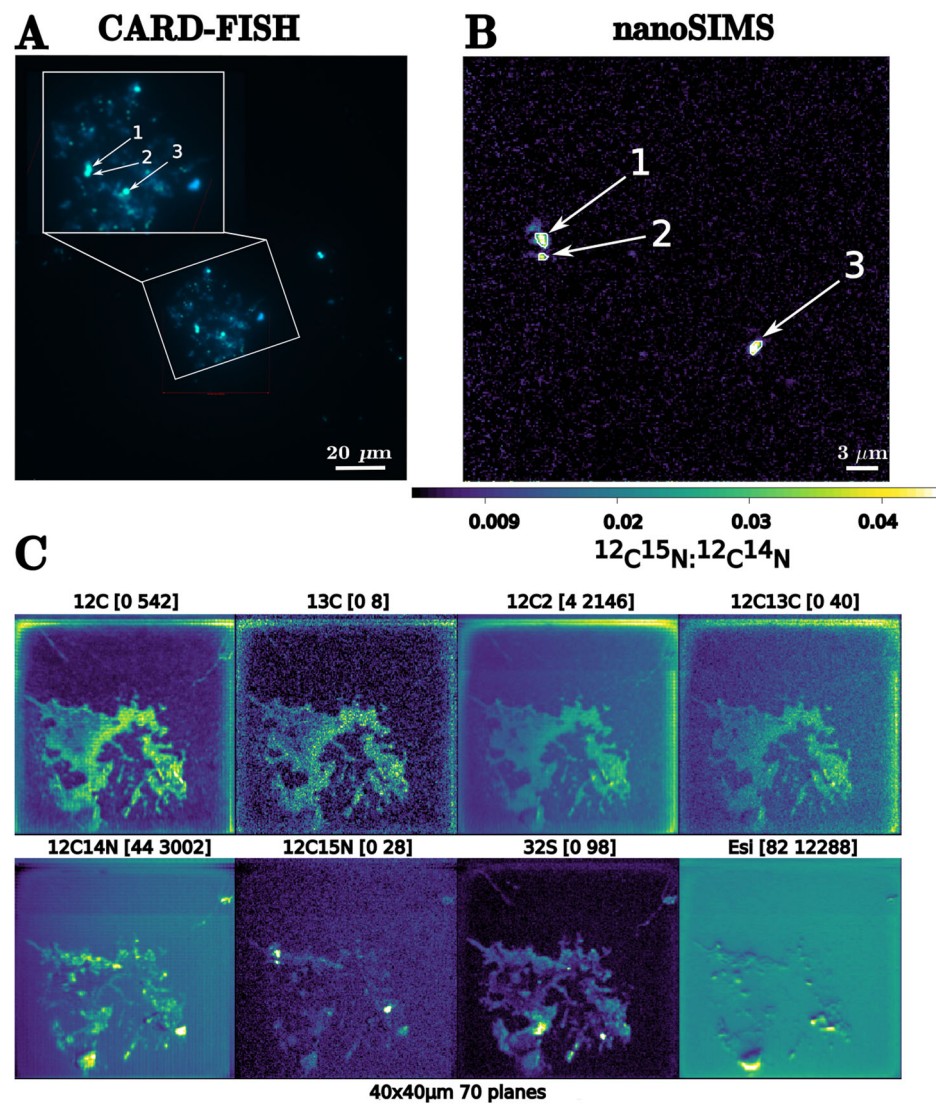

Gammaproteobacteria single-cell $N_2$ fixation rates ranged between $10.17 \pm 5.80$ and $67.46 \pm 48.54$ fmol N cell$^{-1}$ d$^{-1}$ and were highest in the SUSP fraction ($37.33 \pm 13.10$ and $58.14 \pm 63.07$ fmol N cell$^{-1}$ d$^{-1}$), except for station S20 ($10.17 \pm 5.80$ fmol N cell$^{-1}$ d$^{-1}$) where the highest rates were observed in the SS fraction ($67.46 \pm 48.54$ fmol N cell$^{-1}$ d$^{-1}$) (Fig. 2A, Supplementary Data S2). However, a Tukey statistical test showed that Gammaproteobacteria single-cell $N_2$ fixation rates were not significantly different between MSC fractions ($p$ value: SUSP-SS = 0.73). No active $N_2$-fixing Gammaproteobacteria were detected in the FS fraction, thus significance analyses were not done to compare SS-FS and SUSP-FS fractions. Putative NCDs single-cell $N_2$ fixation rates were generally higher than those of Gammaproteobacteria. In the SUSP fraction the highest average rates were $71.27 \pm 61.54$, in the SS fraction at $33.86 \pm 0.65$ and in the FS fraction at $121.44 \pm 22.02$ fmol N cell$^{-1}$ d$^{-1}$ (Fig. 2B, Supplementary Data S3). Contrary to Gammaproteobacteria, active $N_2$ fixation was found for putative NCDs in the FS fraction at station S20. However, putative NCD rates were also not significantly different between MSC fractions (Tukey $p$ value: SUSP-SS = 0.51, SUSP-FS = 0.30, SS-FS = 0.10).

### Diazotroph community composition

Sequencing of the *nifH* gene for different MSC fractions across stations yielded a total of 628 ASVs (Supplementary Data S4), reaching saturation based on the rarefaction index (Fig. S2). Cyanobacteria (Cyanophyceae) were generally the prevalent diazotrophs in the SUSP and SS fractions at all

stations, with a relative abundance ranging between 25% and 90%, generally decreasing from the SUSP to the FS fraction (Fig. 3). S01 to S06 were particularly dominated by cyanobacterial groups (80–90%), while at the remaining stations cyanobacteria were typically <25% of the diazotrophic community (Fig. 3). A redundancy analysis (RDA) showed a positive correlation of S01, S02, S06, and S11 to temperature and oxygen, and a negative correlation to $PO_4^3$, $NO_3^- + NO_2^-$, and $Si(OH)_4$ (Fig. S3). The nitroplast of *Braarudosphaera bigelowii* (*Candidatus* Atelocyanobacterium or UCYN-A (Fig. 3)), represented between 15% and 80% of the total diazotrophic community in all the stations (Fig. 3). *Trichodesmium* and *Crocosphaera* were also detected but were less abundant and more variable across stations and MSC fractions (Fig. 3). Conversely, the relative abundance of NCDs increased up to 75% in the FS fraction. However, NCDs were also detected in the SUSP and SS fractions with relative abundances ranging between 10% and 60% (Fig. 3).

The NCD assemblage was dominated by Gammaproteobacteria, with relative abundances ranging from approximately zero to 20% at stations S01 through S06, and between 20% and 75% at stations S07 through S28 (Fig. 3). The second most abundant NCD group was Betaproteobacteria, with a peaking relative abundance of 20% at station S01. Groups with low relative abundance (below 1%) were pooled together, including Alphaproteobacteria and Deltaproteobacteria (Fig. 3). At stations where single-cell $N_2$ fixation was observed (i.e., S06, S09 and S20), the Gammaproteobacterium *Marinobacter* emerged as the predominant genus representing more than

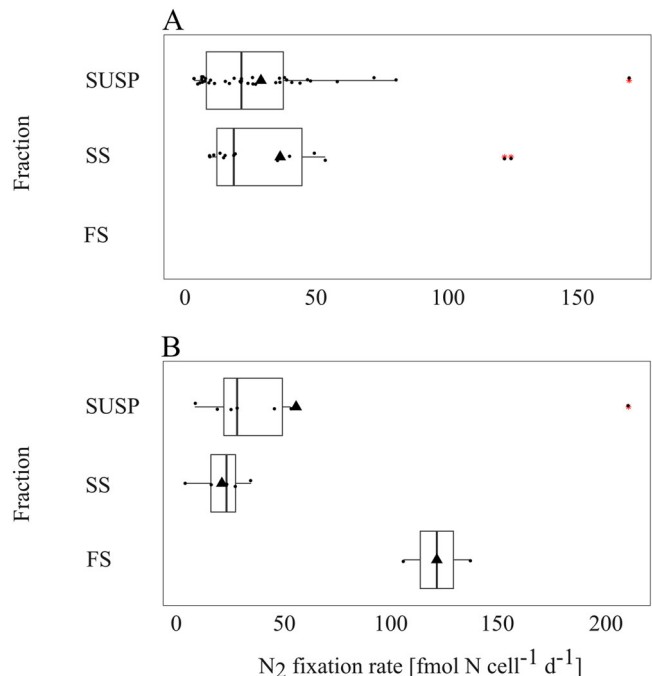

**Fig. 2 | Single-cell N₂ fixation rates in diffrent MSC fractions.** Single-cell N₂ fixation rates of Gammaproteobacteria cells (**A**), and single N₂ fixation rate from putative cells (**B**) across marine snow catcher (MSC) fractions. MSC fractions are identified on the y-axis and N₂ fixation rates on the x-axis. Gammaproteobacteria N₂ fixation rates could not be detected in the FS fraction. N₂ fixation rates are provided in Supplementary Data S5.

50% of the total NCDs assemblage across all MSC fractions. The only exception was the SUSP fraction at station S06 where *Marinobacter* constituted ~35% of the NCD community (Fig. 4). At stations S06 and S09 where temperatures were <15 °C, a higher relative abundance of *Pseudomonas* (up to 25%) was observed, while their relative abundance was below 5% at temperatures between 15 °C and 20 °C at S20. *Klebsiella* was only found at station S06 in the SUSP fraction where it represented roughly 25% of the total NCD community. *Thalassolituus* were primarily found at station S20 in all MSC fractions (decreasing from the SUSP (20%) to FS (5%) fractions). Only at station S09 was *Thalassolituus* observed to represent 15% of the NCD community in the SS fraction. Moreover, the SUSP and FS fractions at station S20 also contained *Oceanobacter* (10% - 25%) (Fig. 4).

## Discussion

Although particles are considered a favorable niche for NCDs, little is known about the N₂ fixation potential of these associations or differences among particle types[58,59]. To the best of our knowledge, only one prior study has demonstrated active N₂ fixation of putative NCDs associated with particles smaller than 210 μm[30]. Our Gammaproteobacteria and putative NCD single-cell N₂ fixation rates are higher than those reported by Harding et al.[30], who measured rates of 0.76 ± 1.6 fmol N cell⁻¹ d⁻¹ from surface waters in the same region, but during the late Fall (November). Additionally, particle-associated NCD N₂ fixation rates of up to ~1 fmol N cell⁻¹ d⁻¹ have been predicted from models[22]. These differences may be associated with different particle size ranges considered among studies, suggesting that full particle size spectrum measurements are needed to assess the relevance of NCDs in pelagic N₂ fixation. The single-cell NCD N₂ fixation rates measured here (Gammaproteobacteria ranging between 10.2–67.5 fmol N cell⁻¹ d⁻¹; Supplementary Data S2 and putative NCD 14.7–121.4 fmol N cell⁻¹ d⁻¹; Supplementary Data S3) are comparable to those of the *B. bigelowii* nitroplast, UCYN-A1, (10–15 fmol N cell⁻¹ d⁻¹)[60], and in the same order of magnitude as those of filamentous cyanobacteria such as *Aphanizomenon*, *Dolichospermum* or *Nodularia* (10–100 fmol N cell⁻¹ d⁻¹)[61], or of *Trichodesmium* (up

to 120 fmol N cell⁻¹ d⁻¹)[62]. Thus, our data suggests that NCD-specific rates are comparable to cell-specific rates of prominent cyanobacterial diazotrophs, despite being measured at the base of the euphotic zone. This highlights the potential of NCDs role in the mesopelagic zone of the North Pacific Gyre. Given that the mesopelagic zone is characterized by limited light and a scarcity of primary producers, the presence of NCDs active on particles indicates that they might play an important role to nitrogen input in the mesopelagic zone of the North Pacific Gyre. From a nitrogen budget perspective, it raises the question of whether NCDs can contribute to the reactive nitrogen stock in the deep chlorophyll maximum and surface layers. However, temperature profiles in Fig. S1 suggest the mixed layer depth was predominantly above or around 50 m for most stations, which suggests that the deep chlorophyll maximum was most likely sustained by vertical nutrient input. To quantify their contribution to bulk pelagic N₂ fixation, knowledge of the abundance of particles of different sizes and the % particles of each size class colonized by active NCDs would be needed. Particle profiles by size class can be obtained, for instance, using laser in-situ scattering and transmissometry (LISST) and underwater vision profiler (UVP)[63–65]. While we deployed both instruments during our cruise and obtained UVP particle profiles successfully (data not shown), the size of the particles with active N₂-fixing NCDs was 29 to 128 μm (Supplementary Data S2), which falls below the detection range of the UVP and thus did not allow a particle profile extrapolation of particle-associated N₂ fixation rates for this study. Active N₂-fixing NCDs have been observed in larger particles (e.g., >210 μm, Harding et al.[30]), suggesting that a combination of LISST and UVP profiles is needed for extrapolation purposes in future studies as the size of particles bearing active NCDs may change among regions, seasons and depths.

Gammaproteobacteria particle-associated N₂ fixation rates were highest in the SUSP fraction (Fig. 2A). SUSP particles have been shown to harbor high oxygen consumption rates that sustain microbial activity in mesopelagic waters[66]. A recent study showed that SUSP particles had a higher carbon content compared to the FS fraction in the Scotia Sea and Benguela currents[66], which is also the case at most stations in our study (Fig. S5). The increased carbon content in the SUSP particles might fulfill the energy demands of Gammaproteobacterial NCDs[59], explaining their high N₂ fixation activity in this fraction compared to others. The NCD community in the SUSP fraction was dominated by *Marinobacter* (ASV-2) which was closely related (>99%) to the environmental genome Tara_IOS_50_MAG_00116[32]. MAG116 has a wide distribution throughout the global ocean, and is particularly abundant in the Indian Ocean[32], suggesting it may contribute to N₂ fixation globally and perhaps also in the sunlit ocean[32]. MAG116 not only contains the full set of genes for N₂ fixation, but also genes for denitrification and assimilatory sulfate reduction[32]. Using BlastKOALA[67], we find that MAG116 has a near complete set of genes for bacterial chemotaxis (ko02030) and flagellar assembly (ko02040), suggesting this organism is geared for a particle-associated lifestyle[28]. However, the relative abundance of *Marinobacter nifH* gene reads had a significant negative correlation with single-cell N₂ fixation rates (Fig. S4). This might reflect potential amplification biases for *Marinobacter*, or perhaps that active N₂ fixation is not limited to highly abundant groups but could also be driven by NCDs present at low abundance. Our nanoSIMS analyses were based on mapping with a general Gammaproteobacteria CARD-FISH probe, so we cannot confirm if the ¹⁵N-enriched Gammaproteobacteria cells detected were *Marinobacter* or also represented other genera. The taxonomy of putative NCDs is unknown (cells not detected by the Gammaproteobacteria CARD-FISH assay but showing significant ¹⁵N enrichment). Regardless, Gammaproteobacteria were by far the group with the highest relative abundances (roughly 50–90%) among the NCDs (Fig. 4), which might also explain the few data points from the putative NCDs (i.e., not abundant). Nevertheless, it should be kept in mind that NCDs in the interior of particles were not measurable, as the nanoSIMS technique does not penetrate the surface of the particles. Thus, this approach may underestimate the particle-associated density of both Gammaproteobacteria and putative NCDs.

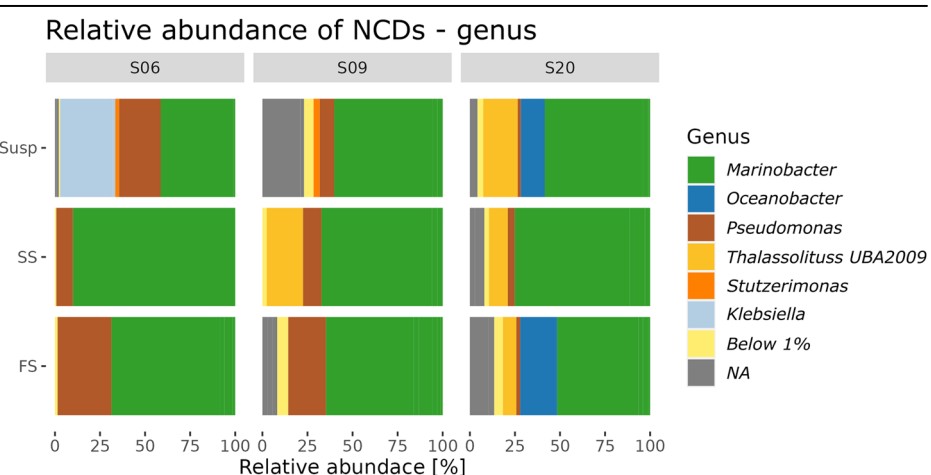

**Fig. 3 | Relative abundance of diazotrophic communities across MSC fractions.** Relative abundance of the diazotrophic community in marine snow catcher (MSC) fractions at each station. MSC fractions are shown in a descending order from SUSP to FS with the diazotrophic community in class (top) and genus (bottom). Amplicon sequence variants (ASVs) below 1% were pooled together and considered as a low abundant species.

**Fig. 4 | Relative abundance of non-cyanobacterial diazotrophs across MSC fractions.** Relative abundance of the NCD community at the genus level from each marine snow catcher (MSC) fraction at stations where ¹⁵N-enriched cells were detected. Amplicon sequence variants (ASVs) related to NCDs were filtered out and normalized for each station and MSC fraction. MSC fractions are in descending order from SUSP to FS. ASVs below 1% relative abundance were pooled. Exact values are provided in Supplementary Data S4. Note all genera belonged to the Gammaproteobacteria Class, except for the NA, which were unidentified sequences.

Different particle fractions (i.e., SUSP, SS, and FS) harbored unique NCD communities dominated by Gammaproteobacteria (~50%), as reported in prior studies in the North Pacific Gyre[9]. The diazotroph community composition reflected a higher relative abundance of NCDs in the FS fraction and a higher relative abundance of cyanobacterial diazotrophs in the SUSP fraction (Fig. 3). This was also accompanied by increased species richness and diversity from the SUSP to the FS fraction (Fig. 4). We further explored the variability of the diazotroph community composition among MSC fractions with a vertical connectivity plot (Fig. 5). Assuming that sinking particles derive from the SUSP fraction[58], we made a vertical connectivity analysis (i.e., the interconnection or exchange of a given diazotrophs between MSC fractions; Fig. 5)[58]. This analysis revealed that SS and FS fractions were largely dominated by ASVs which were also detected in the SUSP fractions (Fig. 5). ASVs uniquely detected in the SS fraction amounted to ~13%, while ASVs only detected in the FS fraction constituted ~40% of

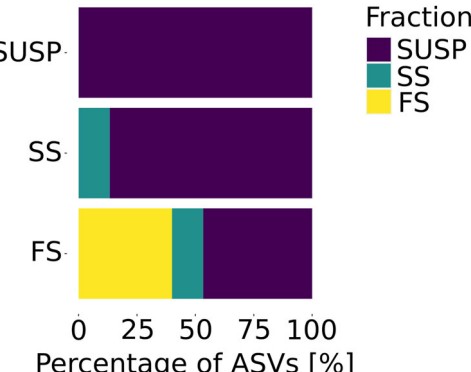

**Fig. 5 | Vertical connectivity of ASVs across MSC fractions.** Vertical connectivity plot across marine snow catcher (MSC) fractions in descending order from SUSP to FS. The plot shows the relative contribution of single Amplicon sequence variants (ASVs). The contributions were determined by averaging the relative abundance over all stations in each fraction. Only ASVs with a contribution above 1% were considered for this analysis. The plot illustration data from all stations analyzed in this study.

the total diazotrophic community (Fig. 5). The high abundance of unique ASVs in the FS fraction might also be reflected in the species richness and diversity indices which were in the FS fraction (Fig. 6). We observed that the SS and FS fractions were largely dominated by ASVs also detected in the SUSP fraction, but ~40% of the ASVs were solely detected in the FS fraction (Fig. 5). This finding is in line with a previous study finding that the prokaryotic community greatly differs between FS and SUSP fractions in the North Atlantic mesopelagic zone[68] and that sinking particle-attached microbes are functionally distinct from their non-sinking (i.e., free-living) counterparts[69]. These results are further in line with previous studies showing how bacterial succession on particles changes as particles sink[58].

Our results support the idea of particles being a favorable habitat for NCDs but also show a difference in the diazotrophic community between particle types (i.e., SUSP, SS, and FS particles). We further support the finding of heterotrophic $N_2$ fixation[30] and provided that NCDs, and particularly Gammaproteobacteria, are globally widespread[32,38] our results further indicates that particle-associated NCD $N_2$ fixation in the ocean might influence particle dynamics in the mesopelagic zone. Diazotrophy-derived ammonium in mesopelagic particles could fuel nitrification[70], or promote particulate organic matter respiration, potentially reducing the efficiency of the biological carbon pump. This could thus counterbalance the enhancement of the biological carbon pump by $N_2$ fixation in surface waters of oligotrophic regions[71].

## Conclusions

We analyzed NCDs associated with different particle fractions and measured their particle-associated $N_2$ fixation rates at 150 m in the North Pacific Ocean. We found that the relative abundance of NCDs was higher in FS particles compared to SUSP particles, with Gammaproteobacteria being the dominant NCDs, particularly *Marinobacter*. We found that the diazotrophic community on the FS fraction greatly differs (40%) from the SUSP and SS fraction, suggesting that the diazotroph community undergoes succession changes as particles sink. NCD single-cell $N_2$ fixation rates were in the same order of magnitude as previous measurements in the surface ocean, suggesting that particle-associated NCDs are important contributors to $N_2$ fixation in the upper mesopelagic ocean. We found particles ranging between 20 and 300 μm to support $N_2$ fixation by NCDs. We further saw

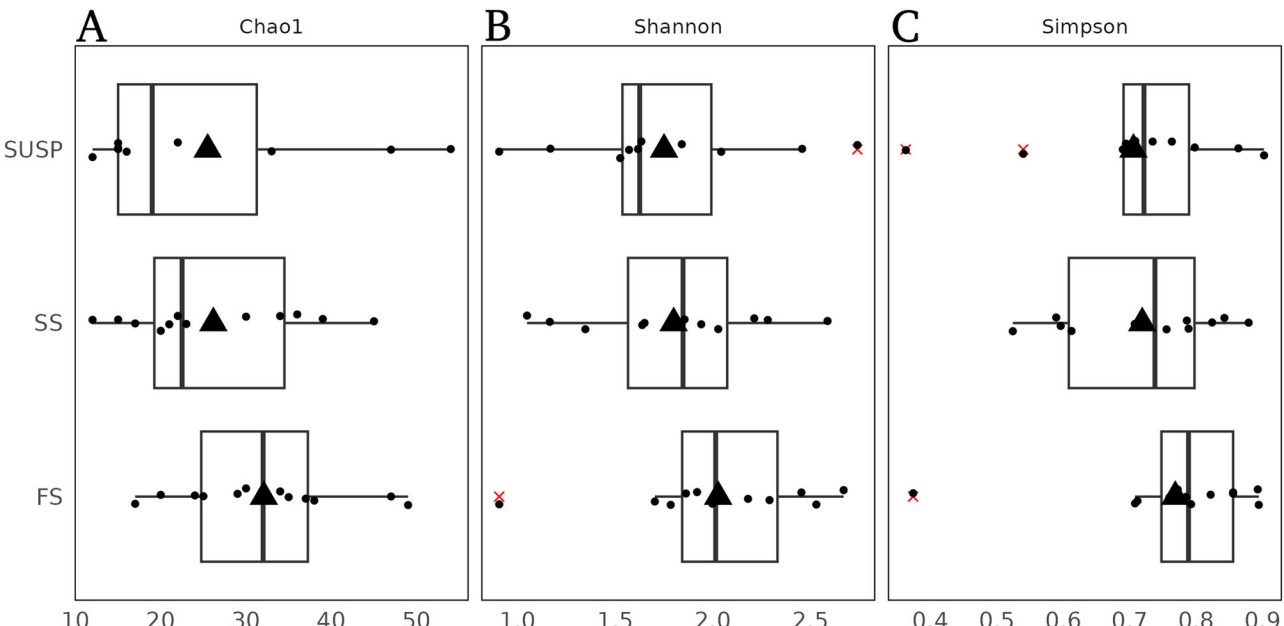

**Fig. 6 | Diversity indices and species richness across MSC fractions.** Boxplot of diversity index and species richness among all marine snow catcher (MSC) fractions at 150 m. Values of Chao1 (**A**), Shannon (**B**), and Simpson (**C**) suggest higher species richness (Chao1) and diversity (Simpson) and evenness (Shannon) in the fast sinking fractions, respectively. Tukey's statistical test did not indicate any significant difference between species richness and diversity.

**Article**

that different particle sizes showed different $N_2$ fixation activity suggesting that the whole particle spectrum needs to be taken into account to quantify the contribution of particle-associated NCD $N_2$ fixation to pelagic nitrogen cycling.

## Data availability

*nifH* gene amplicon sequences are deposited on NCBI under bioproject number PRJNA1085235. The rest of the data presented is available in the supplementary material.

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

## Acknowledgements

This research was funded by an NSF OCE 2023498 grant to K.T.-K., NSF 2023278 grant to K.R.A. at Stanford, BNP-Paribas Foundation for Climate & Biodiversity project NOTION, A*Midex projects MANIOC, and ANITA, and INSU-EC2CO project PANDA granted to M.B. and Horizon MSCA grant 101150634 to C.F.R. Part of this work was performed at the Stanford Nano Shared Facilities (SNSF), supported by the National Science Foundation under award ECCS-2026822. We are indebted to UH's Ocean Technology Group for instrument operations onboard. The authors would also like to thank the captain and crew of the R/V *Kilo Moana* for their help at sea. The contribution of MB was supported by the BIOPOLE National Capability Multicentre Round 2 funding from the Natural Environment Research Council (grant no. NE/W004933/1).

## Author contributions

MB designed the study. M.B. and A.F. conducted sampling at sea. C.F.R. carried out sample and data analysis. K.T.-K. led the cruise and provided *nifH* gene sequencing data. M.M. and C.F.R. did nanoSIMS analysis. A.C. helped with particle sample processing and microscopy. A.V. helped with CARD-FISH analyses. G.V.D., R.C.J., and T.R. assisted with sampling at sea. O.G. runs mass spectrometry analyses. K.R.A. provided comments to the manuscript. L.B. and A.W. helped with the discussion and interpretation of results. C.F.R. wrote the manuscript with input from all co-authors.

## Competing interests

The authors declare no competing interests.
