## [Peer Review file · Communications Biology]

Unveiling the contribution of particle-associated non-cyanobacterial diazotrophs to N₂ fixation in the upper mesopelagic North Pacific Gyre

Corresponding Author: Dr Christian Reeder

Version 0:

Reviewer comments:

Reviewer #1

(Remarks to the Author)

1. Brief summary of the manuscript

The paper describes analysis of diazotroph community composition and N fixation rate on different particle types sampled at 150m depth at a number of stations in the North Pacific, to determine how the community and fixation rate vary with particle sinking rate. There are two primary results: a) greater abundance of NCDs (Non-Cyanobacterial Diazotrophs) on faster sinking particles relative to particles that remain suspended, which the authors interpret as related to growth rate, and b) similarity in cell-specific N fixation rates between particle-bound NCDs and free-living diazotrophs, which they suggest indicates a significant contribution of particle-bound NCD to mesopelagic N fixation.

2. Overall impression of the work

The research is robust, thorough and well-described, although I have a few questions regarding the methodology (see below). The results are clear and reasonably well-presented, and the interpretation sound; consequently, my comments below are mainly questions raised by the research (which after all, is the aim), and that largely require further research to address.

N fixation by particle-bound NCDs is not a new finding as the authors identify by citing publications on particle-attached NCDs and reported N fixation rates. Although not novel the research considerably expands estimates of particle-associated NCDs and N₂ fixation. Further work is required to determine its contribution to marine nitrogen reserves.

The results are of interest to researchers studying nitrogen dynamics, productivity controls and elemental cycling in oligotrophic waters, and are likely to direct further research endeavours in the mesopelagic zone

Although I am not a statistician the statistical analysis appears appropriate. Some details of the methodology are not covered in the text but may well be in referenced papers

3. Specific comments, with recommendations for addressing each comment

Abstract

The results are largely presented in one very long sentence (beginning "Fast-sinking particles showed...") which needs to be rewritten for clarity, probably as two sentences. As written, its unclear whether Gammaproteobacteria are NCDs, and this long sentence switches from fast to slow and then back to fast-sinking particles. This sentence appears to infer there is a difference between NCDs & Gammaproteobacteria N fixation rates, whereas the rates quoted (10-60 fmol N cell.⁻¹ d.⁻¹ and 13-67 fmol N cell.⁻¹ d.⁻¹, respectively) are not significantly different. The latter should be clarified in both Abstract and main text body

Introduction

As in the abstract it should be clarified that Gammaproteobacteria are NCDs

Methods

Unclear in places:

More details are required re sampling from the MSC. Line 110 says "After settling, the SUSP ...fraction" which seems to be an oxymoron. This detail may be covered in the referenced Riley et al paper but it would be useful to explain here
Line 112-113 implies that a filtrate was obtained for N fixation measurement but the following section (Line 121-124) suggests that water was collected for N fixation rate measurement (as there's no mention that the filtrate is resuspended in seawater for N fixation incubation). More detail required.

Why were different volume bottles used for the different size fractions in the N fixation incubation? Potential differences in particle and wall interactions between bottle types

Were the incubation bottles rotated or agitated to maintain the particles in suspension? I appreciate there may be some particle suspension & buoyancy induced by the ship motion, but bearing in mind the significance of the oxygen gradient to N fixation it would seem fairly important to maintain particles in suspension and exclude clumping.

**The formulae on Line 196, 204, 208 and 214 are all missing from the manuscript pdf

Results

It would have been useful to relate the nutrient concentration of the overlying waters to the results, rather than just the sample depth of 150m, as the degree of nitrogen limitation in overlying waters could be a determinant of N fixation rate and community composition. Similarly vertical (depth) distance from the Deep Chlorophyll Maximum could be a major factor influencing particle sinking rate, composition and N fixation

Line 268 indicates that the FS particles had the smallest size-range. It would be interesting to know more regarding particle composition and origin and how this relates to particle size and composition in the overlying deep chlorophyll maximum (assumed to be source of particles).

From Line 367- The results are described in quite granular detail relative to stations. Results would be more accessible and meaningful to readers if the stations could be grouped by type (latitude, nutrient status).

Discussion

Line 345. The comparison with Harding et al result is interesting as the latter reported lower cell specific N fixation in surface waters where nitrogen availability would be assumed to be lower relative to this study where N availability at 150m depth would likely be higher. A further source of variability may be difference in carbon lability between particles in these two studies (see comment above re distance from DCM)

Line 359. It's likely that the 150m sampling depth sits on the nitracline, which is a source of dissolved inorganic nitrogen (DIN) to surface waters via eddy diffusion. This raises the question, why fix nitrogen at these depths when DIN is available (admittedly, not a new question as diazotrophy has been reported at depth and at high DIN previously). From a N budget perspective, how does the N from diazotrophy at 150m contribute to vertical DIN supply to the DCM and overlying surface layer? The latter requires further quantification as identified in Lines 362-364, but these questions should still be considered.
Line 401 "We assume that sinking particles derive from the SUSP fraction". It would have been useful to include further analysis to support this, such as organic analysis of particles

Line 415-416 "particle-associated NCD N₂ fixation in the ocean might influence particle dynamics in the mesopelagic zone" requires explanation or context. This sentence infers that NCD N fixation may alter processing of sinking particles and so their lifetime. Perhaps the authors can expand on this?

Minor comments

Missing words - Line 109

Line 147 I assume "positive" means positively-stained cells ?

Figure 6 legend "High values of Chao1 (A), Shannon (B) and Simpson (C) indicate higher 685 species richness and diversity, respectively". This sentence relates two properties to 3 indices

Line 331 "a given diazotrophs"

Line 359 "Giving" should be "Given"

Line 390 What are "interior" NCDs?

Line 409 "as particles sink." (sink is not plural)

Line 430 "that we need to consider the"

430 "whole particle spectrum need to be taken into account to quantify" should be considered in quantification of

Reviewer #2

(Remarks to the Author)

GENERAL COMMENTS:

Reeder et al. present diazotroph community composition and single-cell nitrogen fixation rates determined from particles collected in the North Pacific upper mesopelagic zone, and segregated into three categories: suspended, slow sinking and fast sinking. The authors combined several state-of-the-art methodologies (15N₂ dissolution method for nitrogen fixation measurements, amplicon sequencing for diazotroph community composition study, CARD-FISH for targeted identification of specific diazotroph group of interest, nanoSIMS for single-cell N₂ fixation rate measurements) to investigate and improve our understanding of particle-associated non-cyanobacterial N₂ fixation activity, in hopes of hypothesize about its relative contribution to bulk pelagic N₂ fixation activity.

The manuscript is overall well written, and the scientific question well introduced. However, the authors should carefully

consider the following general comments to improve the overall quality of the manuscript, facilitate the reader's understanding and interpretation of the study, and more specifically refine the structuring and figure illustration of presented data:

- The authors should provide full and detailed descriptions related to the collection, availability, illustration and interpretability of the presented data
- The authors should revisit statements made which very often rely on a limited sub-set of the available data, and are more than once not supported by the data and analyses performed.
- The authors should consider adjusting the manuscript's structuring and move a few paragraphs as suggested below.

SPECIFIC COMMENTS

Lines 30-35: please rephrase sentence for better clarity.

Lines 143-146: paragraph does not belong in section 2.3, but rather in the beginning of section 2.4.

Lines 181-183: please provide more details to justify the selection of stations S06, S07, S09, S20, and S24 for nanoSIMS analyses. The large majority of the manuscript is based on the interpretation of observations made specifically on these selected sites (and even less so when thereafter data from stations S07 and S24 are excluded).

Please highlight these five stations in Figure S1 to illustrate the wide range of biogeochemical conditions that the authors want to highlight, and explain in the text what about the relative abundance of NCDs did support the choice of these particular stations for further investigation.

Line 252: there is no station S08 shown on the sampling maps in Figure S1 A and C, nor in Table S1. The authors include however mentions of this sampling site in some descriptive sections but not in others. Why?

Line 255: there is no station S27 shown on the sampling maps, nor on the Chl A and Temperature profiles in Figure S1 A to D, or on Table S1. Why?

Line 269: Tables S2 and S3 are missing data for two of the five stations selected above for deeper investigation, namely S07 and S24. Please clearly state why that is to not leave the reader guessing.

Line 281: No Gammaproteobacteria single-cell N₂ fixation rates were detected in the FS. This should be stated in the legend of Figure 2 and the data point should be removed.

Additionally, the authors should make it clear for the reader in each of the figures' legend, which stations' data are being illustrated. This should be the case for Figures 1, 2, 5, 6, and Figures S4 and S5.

Line 286: "Putative NCDs single-cell N₂ fixation rates were GENERALLY higher than those of Gammaproteobacteria", not always as it was not the case in the SUSP fraction of station S20 and SS fraction of stations S09, S20

The limited number of stations (3 out five for nanoSIMS investigation, and 3 out of 14 sampled, including S08 and S27) does not allow for comparison between different biogeochemical conditions (or geographical location).

Lines 303-308: Diazotroph community composition descriptions in the text do not match data presented in Figure 3, which is the only one being referenced. Please edit accordingly to respond to the following:

- *Braarudosphaera bigelowii* is not illustrated in Figure 3, nor mentioned in Table S4 showing the Normalised ASV counts and taxonomy. Please clearly refer to *Candidatus Atelocyanobacterium* and indicate UCYN-A alongside, in the text as well as in Figure 3, for more clarity.

- Edit Figure 3 to clearly illustrate for the reader which classes and genera of diazotrophs belong to the main diazotrophs groups of interest presented and discussed here (i.e., cyanobacteria, NCDs, Gammaproteobacteria).

Lines 311-312: Observation made here only exclude stations S07 and S09, explain why?

Lines 326-337: these two paragraphs would better placed in the discussion section

Lines 327: "Oceanobacter (10% - 25%) statistically positively correlated with temperature" Figure S4 does not support this statement; and therefore, the follow up conclusion "thus suggesting that these NCDs thrived in higher temperature" is incorrect and should be adapted or deleted.

Moreover, Figure S4 shows no positive correlation between temperature and any of the NDCs distribution.

Finally, in the legend of Figure S4 aside from which station data was used, specify that data from all three fractions of MSC were used to produce the figure.

Line 337:

- in Figure 6, please clearly state the meaning of each index Chao1 (richness), Shannon (evenness) and Simpson (diversity)

- in Figure S5, the authors should provide more detailed explanation as to the process of interpreting the graph, either in the Figure legend or in the main body of the text.

Lines 370-371:

- "higher carbon and nitrogen content compared to the FS fraction... the case at most stations in our study." The statement is relevant for carbon content but not for nitrogen in the present study. Please clarify.

- Data from Table S5 could also be plotted (maybe as a bar plot) for better visualization.

Lines 385-387: Argument not backed by any evidence, but also countered by the sentence a few lines down (389) stating putative NCDs are likely not abundant.

Amplicon sequencing relative abundance cannot be directly linked to N₂ fixation activity, especially when such a group as "putative NCDs" is not fully identified and could include a mix of diazotrophs organisms. Plus UCYN-A detachment could not be unambiguously dissociated from the putative NCD group (especially considering that UCYN-A was the most abundant diazotrophs group in most sampling sites).

Lines 401-403: details about Figure 5 presented in lines 330-337 should be included here only, to avoid repetition and for a better structuring of the result/discussion sections.

TECHNICAL CORRECTIONS

Line 121: confirm whether it was 30 or 60 mL sub-samples from MSC fractions that were used to run 15N2 incubations experiments (see also line 116).

Lines 196, 204, 208, 214: the formulas cannot be reviewed nor studied if they are not readable. Please adjust the format accordingly.

Line 242: ASV not previously defined, please indicate for the reader in parentheses "(Amplicon Sequence Variant)"

Lines 287-288: Please clarify what values are being cited, or edit the text, as Table S3 shows that the rates currently cited are "HIGHEST averaged rates" for the three MSC fractions. In Table S3 some lower average rates for each of these fractions are also recorded.

Line 358: Rephrase sentence

Line 374: Specify "The NCD community in the SUSP fraction was dominated by..." for clarity.

Lines 377-378: the correlation is significantly opposite.

Line 388: clearly state in the legend of Figure 4 that all genera (except the ones Below 1% and NA) belonged to the Gammaproteobacteria Class

Lines 423-425: Rephrase sentence for more clarity.

Lines 429-430: Rephrase sentence

Line 623: Edit url for Reference 60 in the Reference list.

Line 65: Provide full details for Reference 65.

Reviewer #3

(Remarks to the Author)

Non-cyanobacterial diazotrophs (NCDs) are known to be widespread at the surface of the oceans and to often occur in (or form) particles. A main hypothesis has been that they mainly contribute to nitrogen fixation when inside those particles. However, few quantifications of their nitrogen fixation rates have been collected within plankton so far, and the current dogma is that they contribute much less to oceanic nitrogen input compared to cyanobacterial diazotrophs. Here, authors describe very high rates of nitrogen fixation by NCDs inside particles of the upper mesopelagic zone (150m) in the Pacific Ocean, changing this view. Of course, only a few dozens of particles were explored in one oceanic region. However, in the context of the high prevalence of NCDs across oceans and seas, these results depict NCDs as potentially highly critical contributors to oceanic nitrogen input.

Overall, results of this manuscript are relevant to our global view of nitrogen fixation in the oceans. They stress the need to search for similar NCD fixation rates in the sunlit oceans as well, and lead to important questions. Could it be that Cyanobacterial diazotrophs are the main contributors in the sunlit ocean and NCDs in the deeper layers? To which extent fixed nitrogen in mesopelagic waters can be fuel plankton activity in the sunlit ocean?

It is very interesting to learn that nitrogen fixation rates are of similar levels between cyanobacterial diazotrophs in the sunlit ocean (decades of quantification in situ and in the lab) and NCDs in particles within the upper layer of the mesopelagic zone. However, this does not mean that "particle-associated NCDs are equally important contributors to pelagic N₂ fixation", as currently suggested in the abstract. Authors did not compare total biomass of those NCDs compared to the cyanobacterial diazotrophs. Again, demonstrating similar rates of NCDs and cyanobacterial diazotrophs when under optimal conditions is already a very important step towards understanding nitrogen fixation input by NCDs, and I commend the authors for providing this insight. However, they cannot conclude equal contribution for the oceans as whole with the currently available data.

Small points:

I wonder why the authors elected not to mention environmental genomes of NCDs for the sunlit oceans in the introduction (most of which are particle-associated), and elected not to link their ASVs to those in the results section? First, those genomes are among the best data we currently have to indicate that NCDs are more abundant compared to cyanobacterial diazotrophs in the sunlit oceans (nifH amplicon alone does not prove that a bacteria can do nitrogen fixation, not to mention the critical amplicon problems). Second, the authors here link the most abundant nifH ASV to one environmental genome, but only in the discussion. Why not look at the details of genomic functions and try to find a link to particle-associated lifestyle (e.g., the chemotaxis covered in the introduction)? And since this environmental genome occurs in multiple oceans, with most of its signal in the Indian Ocean (including signal in >3micron size fraction), this is also an opportunity for the authors to suggest that this bacterial population might contribute to nitrogen fixation at a global scale and possibly also in the sunlit ocean. I find this to be quite interesting and relevant. Finally, are other abundant ASVs also linked to a known culture or environmental genome?

Authors write that the nitroplast-containing *Braarudosphaera bigelowii* represents between 15% and 80% of the data. First, this name is not presented in Figure 3. But most importantly, authors have no direct data suggesting that this eukaryotic species is present in the particles. What they have is nifH from the bacterial symbiont (which might have multiple hosts). While there is a problem mentioning a putative eukaryote in the results, the authors can of course suggest its occurrence inside particles in the discussion instead.

The first section of results is named "Environmental conditions and particle abundance" however there are no words on particle abundance. It is unclear why.

In the abstract, the long sentence in lines 30-35 should be split or re-worded for clarity.

Ln 76: accounting for 50% of what? This sentence is not clear. I assume it is 50% of the signal for all diazotrophs however this need to be clarified.

Ln 78: should be “are often present”

Tom O. Delmont

Version 1:

Reviewer comments:

Reviewer #2

(Remarks to the Author)

The manuscript by Reeder et al. has been revised and greatly improved. Clarifications have been provided to 1) better understand the context in which the study was performed, how and why samples were collected as they were, 2) to more easily and accurately connect observations to interpretations made.

Some important and a few minor points still require some attention:

More important elements to be consider are the following:

- Lines 115-117: The sentence is incomplete, also please clarify how the FS fraction was sampled.

- Lines 363-365: Please rephrase to balance the argument with the fact that N contribution from vertical mixing, which could be several order of magnitude higher than N input via N₂ fixation.

In fact, although mixed layer depths (MLD) were not reported here, the temperature profiles in Figure S1 suggest that the MLD were located above and around 50 m for most stations, which would mean that the deep chlorophyll maxima, located at around 100 m and deeper for most stations, were mainly sustained by vertical nutrient input.

- Line 413: The authors' use, and interpretation of the canonical correspondence analysis is still unclear, especially considering how it can support the fact that the species richness and diversity increased in the study from the SUSP to the FS fraction.

Looking at the legend of Figure S5, the fact that “the relative abundance of ASVs vary across MSC fractions” is already clearly illustrate in Figure 3. What else can be said about Figure S5 and about the fact that MSC fractions explain only 6% of the observed variance. Any hypothesis on what else could contribute to the remaining 94% variance?

- Lines 436-438: This is quite hypothetical and does not account for surface N₂ fixation contributing to the opposite effect which is to fuel the biological carbon pump in nitrogen-depleted surface waters like the ones investigated here. Please rephrase.

Minor details left to be adapted are the following

- Line 264: S27 should be removed like S08

- Lines 706-709: Specify in the legend of Figure 2 that data from stations S06, S09 and S20 were the ones used to produce the two boxplots.

Reviewer #3

(Remarks to the Author)

Authors have addressed my comments, and revisions have improved the overall quality of the study.

Response letter on reviews for manuscript COMMSBIO-24-5974 submitted to Communications Biology

Manuscript title: *'Unveiling the contribution of particle-associated non-cyanobacterial diazotrophs to N₂ fixation in the upper mesopelagic North Pacific Gyre'*

We are very thankful to the editors and reviewers for the suggestions and constructive comments on the global content of the manuscript as well as the specific comments, all certainly allowing us to improve the overall content of this manuscript. Please find below all your comments copied in blue and our response in black font.

Reviewers' comments:

Reviewer #1 (Remarks to the Author):

1. Brief summary of the manuscript

The paper describes analysis of diazotroph community composition and N fixation rate on different particle types sampled at 150m depth at a number of stations in the North Pacific, to determine how the community and fixation rate vary with particle sinking rate. There are two primary results: a) greater abundance of NCDs (Non-Cyanobacterial Diazotrophs) on faster sinking particles relative to particles that remain suspended, which the authors interpret as related to growth rate, and b) similarity in cell-specific N fixation rates between particle-bound NCDs and free-living diazotrophs, which they suggest indicates a significant contribution of particle-bound NCD to mesopelagic N fixation.

2. Overall impression of the work

The research is robust, thorough and well-described, although I have a few questions regarding the methodology (see below). The results are clear and reasonably well-presented, and the interpretation sound; consequently, my comments below are mainly questions raised by the research (which after all, is the aim), and that largely require further research to address.

N fixation by particle-bound NCDs is not a new finding as the authors identify by citing publications on particle-attached NCDs and reported N fixation rates. Although not novel the research considerably expands estimates of particle-associated NCDs and N₂ fixation. Further work is required to determine its contribution to marine nitrogen reserves.

We appreciate this reviewer's comments. Indeed, originally we aimed at estimating the contribution of particle-associated N₂ fixation to bulk N₂ fixation by extrapolating particle-associated rates to particle concentration profiles obtained from a combination of LISST and UVP devices. Unfortunately, the LISST background was unstable during our deployments and we could not use the data further. Although we did obtain UVP profiles successfully, the size of the particles with active N₂-fixing NCDs was 29 to 128 μm (Table S2), which falls below the detection range of the UVP and does not allow a particle profile extrapolation of particle-associated N₂ fixation rates.

We now write in lines 377-385:

“Particle profiles by size class can be obtained, for instance, using laser in-situ scattering and transmissometry (LISST) and underwater vision profiler (UVP) 63-65. While we deployed both instruments during our cruise and obtained UVP particle profiles successfully (data not shown), the size of the particles with active N₂-fixing NCDs was 29 to 128 μm (Table S2), which falls below the detection range of the UVP and thus did not allow a particle profile extrapolation of particle-associated N₂ fixation rates for this study. Active N₂-fixing NCDs have been observed in larger particles (e.g., >210 μm, Harding et al. 2022), suggesting that a combination of LISST and UVP profiles is needed for extrapolation purposes in future studies as the size of particles bearing active NCDs may change among regions, seasons and depths.”

The results are of interest to researchers studying nitrogen dynamics, productivity controls and elemental cycling in oligotrophic waters, and are likely to direct further research endeavours in the mesopelagic zone.

Although I am not a statistician the statistical analysis appears appropriate. Some details of the methodology are not covered in the text but may well be in referenced papers

3. Specific comments, with recommendations for addressing each comment

Abstract

The results are largely presented in one very long sentence (beginning “Fast-sinking particles showed...”) which needs to be rewritten for clarity, probably as two sentences. As written, its unclear whether Gammaproteobacteria are NCDs, and this long sentence switches from fast to slow and then back to fast-sinking particles. This sentence appears to infer there is a difference between NCDs & Gammaproteobacteria N fixation rates, whereas the rates quoted (10-60 fmol N cell.¹ d.¹ and 13-67 fmol N cell.¹ d.¹, respectively) are not significantly different. The latter should be clarified in both Abstract and main text body

We have rewritten the abstract for clarity. Indeed, the Gammaproteobacteria and putative NCDs rates are not significantly different, which was stated in section 3.2, and is now also mentioned in the abstract:

“Dinitrogen (N₂) fixation supports marine life through the supply of reactive nitrogen. Recent studies suggest that particle-associated non-cyanobacterial diazotrophs (NCDs) could contribute significantly to N₂ fixation contrary to the paradigm of diazotrophy as primarily driven by cyanobacterial genera. We examine the community composition of NCDs associated with suspended, slow, and fast-sinking particles in the North Pacific Subtropical Gyre. Suspended and slow-sinking particles showed a higher abundance of cyanobacterial diazotrophs than fast-sinking particles, while fast-sinking particles showed a higher diversity of NCDs including *Marinobacter*, *Oceanobacter* and *Pseudomonas*. Using single-cell mass spectrometry we find that Gammaproteobacteria N₂ fixation rates were higher on suspended and slow-sinking particles (up to 67 ± 48.54 fmol N cell⁻¹ d⁻¹), while putative NCDs’ rates were highest on fast-sinking particles (121 ± 22.02 fmol N cell⁻¹ d⁻¹). These rates are comparable to previous diazotrophic cyanobacteria

observations, suggesting that particle-associated NCDs may be important contributors to pelagic N₂ fixation.”

Introduction

As in the abstract it should be clarified that Gammaproteobacteria are NCDs

It has been clarified in lines 80 to 81:

“Using CARD-FISH combined with single-cell mass spectrometry, we measure specific N₂ fixation rates of Gammaproteobacteria NCDs and other ‘putative NCDs’ ”.

Methods

Unclear in places:

More details are required re sampling from the MSC. Line 110 says “After settling, the SUSP ...fraction” which seems to be an oxymoron. This detail may be covered in the referenced Riley et al paper but it would be useful to explain here

We now write in lines 113-114:

“After the 4 h period, the SUSP and SS fractions were collected...”.

Line 112-113 implies that a filtrate was obtained for N fixation measurement but the following section (Line 121-124) suggests that water was collected for N fixation rate measurement (as there is no mention that the filtrate is resuspended in seawater for N fixation incubation). More detail required.

We apologize for the confusion. From replicate samples, 3 replicates are filtered for downstream DNA analyses, while the other 3 are incubated with ¹⁵N₂ for N₂ fixation measurements. We now clarify this in lines 115-119 as follows:

“For DNA analyses, the SUSP and SS fractions were collected using acid-clean tubing in triplicate polycarbonate bottles of 4500 ml and 500 ml, respectively (Nalgene, Rochester, NY, USA). The FS fraction was collected using sterile and split in triplicate 30 ml subsamples.

2.2 ¹⁵N₂ incubations

Triplicate sub-samples from each MSC fraction (4500 ml, 500 ml and 60 ml from the SUSP, SS, and FS fractions respectively) were collected in transparent polycarbonate bottles with Teflon-coated septum screw-caps (Nalgene, Waltham, MA, USA) as described above, and spiked with 10% v/v ¹⁵N₂-enriched filtered seawater (Cambridge Isotopes Inc., Tewksbury, MA, USA) according to the dissolution method as described in White et al. 2020³⁶. The volumes incubated for each fraction vary as a result of the differences in volume available from the MSC fractions. ”

Why were different volume bottles used for the different size fractions in the N fixation incubation? Potential differences in particle and wall interactions between bottle types

Each MSC fraction has a different volume (SUSP = 93 L , SS = 5 L, FS = 2 L), so it is unfortunately not possible to incubate equal volumes for each fraction considering POC/PON concentrations typical of each fraction (Table S5) and analytical detectability limitations. The volume of dissolved $^{15}\text{N}_2$ (2.5-4.5 N atom%) added for incubations is adapted to each MSC fraction, and all fractions are incubated under the same conditions. The material of the incubation bottles was the same for the SUSP and SS fractions (polycarbonate), but other containers had to be used for the smaller FS fraction due to the unavailability of suitable septum capped polycarbonate commercial options for 30 ml incubations.

Were the incubation bottles rotated or agitated to maintain the particles in suspension? I appreciate there may be some particle suspension & buoyancy induced by the ship motion, but bearing in mind the significance of the oxygen gradient to N fixation it would seem fairly important to maintain particles in suspension and exclude clumping.

No, they were in the dark in a cold room. The only agitation was the ship's movement. Considering the low particle load in this oligotrophic system and the small size of the particles found (5-246 μm range), we assume that aggregation or 'clumping' was negligible in a 24 h incubation period.

**The formulae on Line 196, 204, 208 and 214 are all missing from the manuscript pdf

We apologize for this mistake. Formulae have been added in the revised version.

Results

It would have been useful to relate the nutrient concentration of the overlying waters to the results, rather than just the sample depth of 150m, as the degree of nitrogen limitation in overlying waters could be a determinant of N fixation rate and community composition. Similarly vertical (depth) distance from the Deep Chlorophyll Maximum could be a major factor influencing particle sinking rate, composition and N fixation

The upper water column nutrient concentrations will be reported in other companion publications, currently in preparation by the UCSC team. Figure S1 shows chlA from surface to 150 m. We thus sampled at roughly 25-75 m from the DCM. We have looked into the distance between the DCM and 150 m at each station and compared it to POC and PON concentrations in each MSC fraction (see figure below), but no significant relationships were found.

Line 268 indicates that the FS particles had the smallest size-range. It would be interesting to know more regarding particle composition and origin and how this relates to particle size and composition in the overlying deep chlorophyll maximum (assumed to be source of particles).

We certainly agree that further biochemical information would provide insights into the particle origin. Unfortunately, all that we can provide in that sense is POC/PON concentrations, which can be found on Figure S6.

From Line 367- The results are described in quite granular detail relative to stations. Results would be more accessible and meaningful to readers if the stations could be grouped by type (latitude, nutrient status).

We certainly agree it would be useful to discuss results in a biogeochemical/geographical perspective. However, as another reviewer points out, the limited number of stations (3 out of five for nanoSIMS investigation, and 3 out of 14 sampled) does not allow for comparison between different biogeochemical conditions (or geographical location).

Discussion

Line 345. The comparison with Harding et al result is interesting as the latter reported lower cell specific N fixation in surface waters where nitrogen availability would be assumed to be lower relative to this study where N availability at 150m depth would likely be higher. A further source of variability may be difference in carbon lability between particles in these two studies (see comment above re distance from DCM)

Indeed, active N₂ fixation in the mesopelagic and other nitrogen-rich environments continues to puzzle the scientific community. As discussed by other colleagues, this may be explained by transient low nitrogen concentrations within the particle (Chakraborty et al., 2021), or alternatively N₂ fixation may be used as a shunt for excess energy and not to obtain reactive nitrogen (Bombar et al., 2016). The surface particles collected in Harding et al. were likely more labile than the ones we collected at 150 m depth. However, their sampling approach (concentration of large particles with a 210 µm mesh net) is very different from ours and we fear the types and sizes of particles collected are not directly intercomparable. Nevertheless, the Harding et al. study is the only other single-cell particle-associated N₂ fixation study published to date, so we need to use it for comparison.

Line 359. Its likely that the 150m sampling depth sits on the nitracline, which is a source of dissolved inorganic nitrogen (DIN) to surface waters via eddy diffusion. This raises the question, why fix nitrogen at these depths when DIN is available (admittedly, not a new question as diazotrophy has been reported at depth and at high DIN previously). From a N budget perspective, how does the N from diazotrophy at 150m contribute to vertical DIN supply to the DCM and overlying surface layer? The latter requires further quantification as identified in Lines 362-364, but these questions should still be considered.

Please refer to the previous comment for discussion on active N₂ fixation in reactive nitrogen rich waters. The possibility of mesopelagic N₂ fixation replenishing reactive nitrogen stocks in the upper water column is indeed a good point. We have added to lines 371-373.

“Furthermore, from a N budget perspective, it raises the question of whether NCDs can contribute with dissolved inorganic nitrogen to the deep chlorophyll maximum and surface layer”

Line 401 “We assume that sinking particles derive from the SUSP fraction”. It would have been useful to include further analysis to support this, such as organic analysis of particles

POC and PON concentration from each MSC fraction are provided in Figure S6. We refer to this figure in line 391.

Line 415-416 “particle-associated NCD N₂ fixation in the ocean might influence particle dynamics in the mesopelagic zone” requires explanation or context. This sentence infers that NCD N fixation may alter processing of sinking particles and so their lifetime. Perhaps the authors can expand on this?

Indeed this merits discussion, but unfortunately it comes down to mere speculation at present. We have added in lines 447-449:

“ Diazotrophy-derived ammonium in mesopelagic particles could fuel nitrification⁷⁰, or promote particulate organic matter respiration, potentially reducing the efficiency of the biological carbon pump.”.

Minor comments

Missing words - Line 109

Corrected - line 112-113:

“with the modification that particles were settled for 4 h and not 2 h.”

Line 147 I assume “positive” means positively-stained cells ?

Yes - Corrected - line 158-160:

“ CARD-FISH positively stained cells were then mapped for single-cell isotope ratio measurements using nanoSIMS (see section 2.4.) ”

Figure 6 legend “High values of Chao1 (A), Shannon (B) and Simpson (C) indicate higher

Corrected to:

“... Values of Chao1 (A), Shannon (B) and Simpson (C) suggest higher species richness (Chao1) and diversity (Simpson) and evenness (Shannon) in the fast sinking fractions, respectively. Tukey's statistical test did not indicate any significant difference between species richness and diversity.”

685 species richness and diversity, respectively”. This sentence relates two properties to 3 indices

Corrected, please see above.

Line 331 “a given diazotrophs”

Corrected - Line 427:

"A given diazotroph".

Line 359 "Giving" should be "Given"

Corrected - Line 368.

Line 390 What are "interior" NCDs?

Changed to - line 414:

"..that NCDs in the interior of particles were not measurable".

Line 409 "as particles sink." (sink is not plural)

Corrected.

Line 430 "that we need to consider the"

Corrected - lines 462-465:

"We further saw that different particle sizes showed different N₂ fixation activity suggesting that the whole particle spectrum needs to be taken into account to quantify the contribution of particle- associated NCD N₂ fixation to pelagic nitrogen cycling. "

430 "whole particle spectrum need to be taken into account to quantify" should be considered in quantification of

The sentence has been rephrased, please see above.

Reviewer #2 (Remarks to the Author):

GENERAL COMMENTS:

Reeder et al. present diazotroph community composition and single-cell nitrogen fixation rates determined from particles collected in the North Pacific upper mesopelagic zone, and segregated into three categories: suspended, slow sinking and fast sinking. The authors combined several state-of-the-art methodologies ($^{15}\text{N}_2$ dissolution method for nitrogen fixation measurements, amplicon sequencing for diazotroph community composition study, CARD-FISH for targeted identification of specific diazotroph group of interest, nanoSIMS for single-cell N_2 fixation rate measurements) to investigate and improve our understanding of particle-associated non-cyanobacterial N_2 fixation activity, in hopes of hypothesize about its relative contribution to bulk pelagic N_2 fixation activity.

The manuscript is overall well written, and the scientific question well introduced. However, the authors should carefully consider the following general comments to improve the overall quality of the manuscript, facilitate the reader's understanding and interpretation of the study, and more specifically refine the structuring and figure illustration of presented data:

- The authors should provide full and detailed descriptions related to the collection, availability, illustration and interpretability of the presented data
- The authors should revisit statements made which very often rely on a limited sub-set of the available data, and are more than once not supported by the data and analyses performed.
- The authors should consider adjusting the manuscript's structuring and move a few paragraphs as suggested below.

SPECIFIC COMMENTS

Lines 30-35: please rephrase sentence for better clarity.

As reviewer 1 also suggested, below is the corrected abstract:

“Dinitrogen (N_2) fixation supports marine life through the supply of reactive nitrogen. Recent studies suggest that particle-associated non-cyanobacterial diazotrophs (NCDs) could contribute significantly to N_2 fixation contrary to the paradigm of diazotrophy as primarily driven by cyanobacterial genera. We examine the community composition of NCDs associated with suspended, slow, and fast-sinking particles in the North Pacific Subtropical Gyre. Suspended and slow-sinking particles showed a higher abundance of cyanobacterial diazotrophs than fast-sinking particles, while fast-sinking particles showed a higher diversity of NCDs including *Marinobacter*, *Oceanobacter* and *Pseudomonas*. Using single-cell mass spectrometry we find that Gammaproteobacteria N_2 fixation rates were higher on suspended and slow-sinking particles (up to 67 ± 48.54 fmol N cell⁻¹ d⁻¹),

while putative NCDs' rates were highest on fast-sinking particles (121 ± 22.02 fmol N cell⁻¹ d⁻¹). These rates are comparable to previous diazotrophic cyanobacteria observations, suggesting that particle-associated NCDs may be important contributors to pelagic N₂ fixation.”

Lines 143-146: paragraph does not belong in section 2.3, but rather in the beginning of section 2.4.

We agree, but we had to mention this in section 2.3. because CARD-FISH and nanoSIMS are done on the same filters. For clarity, we have added to lines 156-160:

“fixed with 16% microscopy grade paraformaldehyde (1.6% final concentration) and stored at -80°C. These filters were used to identify Gammaproteobacteria cells using a CARD-FISH assay (see below). CARD-FISH positively stained cells were then mapped for single-cell isotope ratio measurements using nanoSIMS (see section 2.4.). ”

Lines 181-183: please provide more details to justify the selection of stations S06, S07, S09, S20, and S24 for nanoSIMS analyses. The large majority of the manuscript is based on the interpretation of observations made specifically on these selected sites (and even less so when thereafter data from stations S07 and S24 are excluded).

We have clarified this selection in:

Section 2.4 - Line 193-195: “*Stations used for nanoSIMS analyses (S06, S07, S09, S20 and S24) were selected based on their spatial location (Fig. S1) to cover a wide range of biogeochemical conditions, and based on the relative abundance of NCDs in each MSC fraction (Fig. 3).*”

Section 3.2 - Lines 289-290: “*We did not find any particles with N₂ fixing cells at stations S07 and S24. Hence, we only refer to stations S06, S09 and S20 from now on.*”

Please highlight these five stations in Figure S1 to illustrate the wide range of biogeochemical conditions that the authors want to highlight, and explain in the text what about the relative abundance of NCDs did support the choice of these particular stations for further investigation.

Please see the above corrections. Fig. S1 has been edited to highlight these five stations, and updated figure legend below:

Figure S1 Sampling stations during the NCD cruise (KM2206). A and B show surface and water column chlorophyll *a* (Chl *a*) concentrations, respectively, C and D show surface and water column temperature, respectively. The data shown is a composite of the cruise duration (i.e., 4 June to 6 July 2022). The data in A and C were obtained from the E.U. Copernicus Marine Service information; DOI 10.48670/moi-00016 and DOI 10.48670/moi-00015. Stations in our cruise track are shown in the color red (stars and circles). Specific stations selected for nanoSIMS are shown as a star.

Line 252: there is no station S08 shown on the sampling maps in Figure S1 A and C, nor in Table S1. The authors include however mentions of this sampling site in some descriptive sections but not in others. Why?

S08 was part of the overall cruise, but not sampled for our analyses. We have thus removed it from former line 266.

Line 255: there is no station S27 shown on the sampling maps, nor on the Chl A and Temperature profiles in Figure S1 A to D, or on Table S1. Why?

Same as for S08. We have removed mention of this station from former line 266.

Line 269: Tables S2 and S3 are missing data for two of the five stations selected above for deeper investigation, namely S07 and S24. Please clearly state why that is to not leave the reader guessing.

These data have been added to table S2 and S3.

Line 281: No Gammaproteobacteria single-cell N₂ fixation rates were detected in the FS. This should be stated in the legend of Figure 2 and the data point should be removed.

Figure 2 has been redrawn and the legend changed as follows:

“Figure 2: Single-cell N₂ fixation rates of Gammaproteobacteria cells (A), and single N₂ fixation rate from putative cells (B) across marine snow catcher (MSC) fractions. MSC fractions are identified on the y-axis and N₂ fixation rates on the x-axis. Gammaproteobacteria N₂ fixation rates could not be detected in the FS fraction.”

Additionally, the authors should make it clear for the reader in each of the figures' legend,

which stations' data are being illustrated. This should be the case for Figures 1, 2, 5, 6, and Figures S4 and S5.

The stations included in each figure have been listed in each corresponding legend.

Line 286: "Putative NCDs single-cell N₂ fixation rates were GENERALLY higher than those of Gammaproteobacteria", not always as it was not the case in the SUSP fraction of station S20 and SS fraction of stations S09, S20

It has been corrected.

The limited number of stations (3 out five for nanoSIMS investigation, and 3 out of 14 sampled, including S08 and S27) does not allow for comparison between different biogeochemical conditions (or geographical location).

We apologize for the confusion. Stations S08 and S27 were not part of our sampling, but of the general cruise transect. These stations will be removed from the figures. We agree that the dataset is too scarce to discuss biogeochemical/geographical variability, and hence present and discuss our results on a station by station basis.

Lines 303-308: Diazotroph community composition descriptions in the text do not match data presented in Figure 3, which is the only one being referenced. Please edit accordingly to respond to the following:

- *Braarudosphaera bigelowii* is not illustrated in Figure 3, nor mentioned in Table S4 showing the Normalised ASV counts and taxonomy. Please clearly refer to *Candidatus Atelocyanobacterium* and indicate UCYN-A alongside, in the text as well as in Figure 3, for more clarity.
- Edit Figure 3 to clearly illustrate for the reader which classes and genera of diazotrophs belong to the main diazotrophs groups of interest presented and discussed here (i.e., cyanobacteria, NCDs, Gammaproteobacteria).

Corrected in lines 321-327:

"The nitroplast of *Braarudosphaera bigelowii* (*Candidatus Atelocyanobacterium* or UCYN-A (Fig. 3)), represented between 15% and 80% of the total diazotrophic community in all the stations (Fig. 3). *Trichodesmium* and *Crocospaera* were also detected but were less abundant and more variable across stations and MSC fractions (Fig. 3). Conversely, the relative abundance of NCDs increased up to 75% in the FS fraction. However, NCDs were also detected in the SUSP and SS fractions with relative abundances ranging between 10% and 60% (Fig. 3)"

And updated in Figure 3:

Lines 311-312: Observation made here only exclude stations S07 and S09, explain why?

Apologize for the confusion. This was an unfortunate typo, as it should be S01 through S06 and S07 through S28. Looking at Figure 3, we also see Gammaproteobacteria ranging between 20 and 75% in S07 and S09. This has been corrected in line 329-331:

“The NCD assemblage was dominated by Gammaproteobacteria, with relative abundances ranging from approximately zero to 20% at stations S01 through S06, and between 20% to 75% at stations S07 through S28 (Fig. 3).”

Lines 326-337: these two paragraphs would better placed in the discussion section

Paragraph regarding Oceanobacter (see next comment) has been deleted. The remaining paragraph (former line 330-337) has been moved to discussion at line 425-433.

Lines 327: “Oceanobacter (10% - 25%) statistically positively correlated with temperature” Figure S4 does not support this statement; and therefore, the follow up conclusion “thus suggesting that these NCDs thrived in higher temperature” is incorrect and should be adapted or deleted.

This has been deleted.

Moreover, Figure S4 shows no positive correlation between temperature and any of the NDCs distribution.

We agree. That statement has been deleted.

Finally, in the legend of Figure S4 aside from which station data was used, specify that data from all three fractions of MSC were used to produce the figure.

This has been added to Fig. S4 caption.

Line 337: - in Figure 6, please clearly state the meaning of each index Chao1 (richness), Shannon (evenness) and Simpson (diversity)
- in Figure S5, the authors should provide more detailed explanation as to the process of interpreting the graph, either in the Figure legend or in the main body of the text.

Corrected - see updated figure legends:

“Figure 6: Boxplot of diversity index and species richness among all marine snow catcher (MSC) fractions at 150 m. High values of Chao1 (A), Shannon (B) and Simpson (C) suggest higher species richness (Chao1), diversity (Simpson) and evenness (Shannon) in the fast sinking fractions. Tukey's statistical test did not indicate any significant difference between species richness and diversity.”

“Figure S5: Canonical Correspondence Analysis (CCA) of amplicon sequence variants (ASVs) from the marine snow catcher (MSC) particle fractions. CCA reveals the relationships between ASV composition and how they are impacted by the different MSC fractions. The x-axis shows CCA1 with 3.4% of variance explained in a constrained space, while 58.1% was explained in an unconstrained space. The y-axis shows CCA2 with 2.4% of the variance explained in a constrained space, while 41.9% was explained in an unconstrained space. The clustering of SUSP, SS, and FS into three separate clusters indicates that the relative abundance of ASVs varies across different MSC fractions. However, it is important to clarify that MSC fractions explain only about 6% of the observed variance.”

Lines 370-371:

- “higher carbon and nitrogen content compared to the FS fraction... the case at most stations in our study.” The statement is relevant for carbon content but not for nitrogen in the present study. Please clarify.

We have changed it to refer to carbon only - Line 390.

- Data from Table S5 could also be plotted (maybe as a bar plot) for better visualization.

Data from Table S5 has been plotted and named Fig. S6. Hence, Table S5 has been removed.

Lines 385-387: Argument not backed by any evidence, but also countered by the sentence a few lines down (389) stating putative NCDs are likely not abundant.

Amplicon sequencing relative abundance cannot be directly linked to N₂ fixation activity, especially when such a group as “putative NCDs” is not fully identified and could include a mix of diazotrophs organisms. Plus UCYN-A detachment could not be unambiguously

dissociated from the putative NCD group (especially considering that UCYN-A was the most abundant diazotrophs group in most sampling sites).

We agree. We have removed this argument.

Lines 401-403: details about Figure 5 presented in lines 330-337 should be included here only, to avoid repetition and for a better structuring of the result/discussion sections.

Agreed. We have moved the text in former lines 330-337 to the discussion section lines 425-433.

TECHNICAL CORRECTIONS

Line 121: confirm whether it was 30 or 60 mL sub-samples from MSC fractions that were used to run 15N2 incubations experiments (see also line 116).

It was 60 mL. We apologize for the confusion. It has now been corrected in the text.

Lines 196, 204, 208, 214: the formulas cannot be reviewed nor studied if they are not readable. Please adjust the format accordingly.

We apologize for this. The formulae have been introduced correctly in the new version.

Line 242: ASV not previously defined, please indicate for the reader in parentheses “(Amplicon Sequence Variant)”

Corrected in line 257:

“...most abundant ASVs (Amplicon Sequence Variants) using...”

Lines 287-288: Please clarify what values are being cited, or edit the text, as Table S3 shows that the rates currently cited are “HIGHEST averaged rates” for the three MSC fractions. In Table S3 some lower average rates for each of these fractions are also recorded.

It is indeed the highest average rate shown. This has been clarified in line 305-306

“ the SUSP fraction the highest average rates were 71.27 ± 61.54 , in the SS fraction at 33.86 ± 0.65 and in the FS fraction at 121.44 ± 22.02 $\text{fmol N cell}^{-1} \text{d}^{-1}$ (Fig. 2B, Table S3). “

Line 358: Rephrase sentence

Corrected in line 367-368:

“This highlights the potential of NCDs role in the mesopelagic zone of the North Pacific Gyre. “

Line 374: Specify “The NCD community in the SUSP fraction was dominated by...” for clarity.

Corrected in line 394-395:

“The NCD community in the SUSP fraction was dominated by Marinobacter (ASV-2)”

Lines 377-378: the correlation is significantly opposite.

Indeed, it has been corrected in lines 391-393:

“However, the relative abundance of Marinobacter nifH gene reads had a significant negative correlation with single-cell N₂ fixation rates (Fig. S4).”

Line 388: clearly state in the legend of Figure 4 that all genera (except the ones Below 1% and NA) belonged to the Gammaproteobacteria Class

Now added to the legend of Figure 4:

“Note all genera belonged to the Gammaproteobacteria Class, except for the NA, which were unidentified sequences. “

Please note that ASV below 1% is all sequences below 1 % relative abundance pooled.

Lines 423-425: Rephrase sentence for more clarity.

Corrected in lines 456-459:

“We found that the diazotrophic community on the FS fraction greatly differs (40%) from the SUSP and SS fraction, suggesting that the diazotroph community undergoes succession changes as particles sink. “

Furthermore, please see lines 414-427 for the discussion on this matter.

Lines 429-430: Rephrase sentence

Corrected in lines 462-465:

“We further saw that different particle sizes showed different N₂ fixation activity suggesting that the whole particle spectrum needs to be taken into account to quantify the contribution of particle-associated NCD N₂ fixation to pelagic nitrogen cycling.”

Line 623: Edit url for Reference 60 in the Reference list.

It has been corrected.

Line 65: Provide full details for Reference 65.

It has been corrected.

Reviewer #3 (Remarks to the Author):

Non-cyanobacterial diazotrophs (NCDs) are known to be widespread at the surface of the oceans and to often occur in (or form) particles. A main hypothesis has been that they mainly contribute to nitrogen fixation when inside those particles. However, few quantifications of their nitrogen fixation rates have been collected within plankton so far, and the current dogma is that they contribute much less to oceanic nitrogen input compared to cyanobacterial diazotrophs. Here, authors describe very high rates of nitrogen fixation by NCDs inside particles of the upper mesopelagic zone (150m) in the Pacific Ocean, changing this view. Of course, only a few dozens of particles were explored in one oceanic region. However, in the context of the high prevalence of NCDs across oceans and seas, these results depict NCDs as potentially highly critical contributors to oceanic nitrogen input.

Overall, results of this manuscript are relevant to our global view of nitrogen fixation in the oceans. They stress the need to search for similar NCD fixation rates in the sunlit oceans as well, and lead to important questions. Could it be that Cyanobacterial diazotrophs are the main contributors in the sunlit ocean and NCDs in the deeper layers? To which extent fixed nitrogen in mesopelagic waters can be fuel plankton activity in the sunlit ocean?

It is very interesting to learn that nitrogen fixation rates are of similar levels between cyanobacterial diazotrophs in the sunlit ocean (decades of quantification in situ and in the lab) and NCDs in particles within the upper layer of the mesopelagic zone. However, this does not mean that “particle-associated NCDs are equally important contributors to pelagic N₂ fixation”, as currently suggested in the abstract. Authors did not compare total biomass of those NCDs compared to the cyanobacterial diazotrophs. Again, demonstrating similar rates of NCDs and cyanobacterial diazotrophs when under optimal conditions is already a very important step towards understanding nitrogen fixation input by NCDs, and I commend the authors for providing this insight. However, they cannot conclude equal contribution for the oceans as whole with the currently available data.

We certainly agree that our dataset is too limited to conclude that particle-associated NCDs contribute as much reactive nitrogen than cyanobacterial diazotrophs to the water column. We have thus toned down this statement as follows (also see abstract):

“These rates are comparable to previous diazotrophic cyanobacteria observations, suggesting that particle-associated NCDs may be important contributors to pelagic N₂ fixation.”

Originally we aimed at estimating the contribution of particle-associated N₂ fixation to bulk N₂ fixation by extrapolating particle-associated rates to particle concentration profiles obtained from a combination of LISST and UVP devices.

We now write in lines 377-385:

“Particle profiles by size class can be obtained, for instance, using laser in-situ scattering and transmissometry (LISST) and underwater vision profiler (UVP) 63-65. While we deployed both instruments during our cruise and obtained UVP particle profiles successfully (data not shown), the size of the particles with active N₂-fixing NCDs was 29 to 128 μm (Table S2), which falls below the detection range of the UVP and thus did not allow a particle profile extrapolation of particle-associated N₂ fixation rates for this study. Active N₂-fixing NCDs have been observed in larger particles (e.g., >210 μm, Harding et al. 2022), suggesting that a combination of LISST and UVP profiles is needed for extrapolation purposes in future studies as the size of particles bearing active NCDs may change among regions, seasons and depths.”

Small points:

I wonder why the authors elected not to mention environmental genomes of NCDs for the sunlit oceans in the introduction (most of which are particle-associated), and elected not to link their ASVs to those in the results section? First, those genomes are among the best data we currently have to indicate that NCDs are more abundant compared to cyanobacterial diazotrophs in the sunlit oceans (nifH amplicon alone does not prove that a bacteria can do nitrogen fixation, not to mention the critical amplicon problems).

We certainly agree that the TARA metagenomes should be mentioned and put into context here. We would like to point out that, just as amplicons, the presence or even expression of a nif operon in metagenome/metatranscriptome does not imply active N₂ fixation either. Only ¹⁵N measurements can measure rates.

We now add in lines 73-82:

“NCDs have been widely detected in the 0.2-5 μm, 5-20 μm, 20-180 μm and 180-2,000 μm filter size fractions of the TARA Ocean metagenomes dataset, with approximately 50% of nifH reads in the 0.5-5 μm size-fraction and up to 25% of total nifH reads in the 180 to 2,000 μm size fractions^{32,33}. NCDs’ metagenome-assembled genomes covering the size range 0.8 to 2000 μm have been shown to be more abundant than their cyanobacteria counterparts in the surface ocean³². This does add to the growing evidence that NCDs often present a particle-associated lifestyle, albeit knowledge of their N₂ fixation potential is limited to direct isotope tracing measurements. Using CARD-FISH combined with single-cell mass spectrometry, we measure specific N₂ fixation rates of Gammaproteobacteria NCDs and other ‘putative NCDs’.”

Additionally, we did take into account the MAGs when annotating the nifH seqs. We used the nifH DADA2 database v2.0.5 which contains all nifH seqs from the HBD Mags. From this, we found ASV-2 belonging to Tara_IOS_50_MAG_00116, being the second most abundant group in our dataset, as this reviewer also states below.

Second, the authors here link the most abundant nifH ASV to one environmental genome, but only in the discussion. Why not look at the details of genomic functions and try to find a link to particle-associated lifestyle (e.g., the chemotaxis covered in the introduction)?

And since this environmental genome occurs in multiple oceans, with most of its signal in the Indian Ocean (including signal in >3micron size fraction), this is also an opportunity for the authors to suggest that this bacterial population might contribute to nitrogen fixation at a global scale and possibly also in the sunlit ocean. I find this to be quite interesting and relevant. Finally, are other abundant ASVs also linked to a known culture or environmental genome?

This is a great point. We added the following lines (396-402) to the discussion. Furthermore, below is a figure for nearly completed pathways for chemotaxis, as well as flagellar assembly, based on Hallstrøm et al. (2022).

“MAG116 has a wide distribution throughout the global ocean, and is particularly abundant in the Indian Ocean ³², suggesting it may contribute to N₂ fixation globally and perhaps also in the sunlit ocean ³². MAG116 not only contains the full set of genes for N₂ fixation, but also genes for denitrification and assimilatory sulfate reduction ³². Using BlastKOALA ⁶⁷, we find that MAG116 has a near complete set of genes for bacterial chemotaxis (ko02030) and flagellar assembly (ko02040), suggesting this organism is geared for a particle-associated lifestyle ²⁸.”

FLAGELLAR ASSEMBLY

02040 11/25/20
(c) Kanehisa Laboratories

BACTERIAL CHEMOTAXIS

02030 10/17/17
(c) Kanehisa Laboratories

Authors write that the nitroplast-containing *Braarudosphaera bigelowii* represents between 15% and 80% of the data. First, this name is not presented in Figure 3. But most importantly, authors have no direct data suggesting that this eukaryotic species is present in the particles. What they have is *nifH* from the bacterial symbiont (which might have multiple hosts). While there is a problem mentioning a putative eukaryote in the results, the authors can of course suggest its occurrence inside particles in the discussion instead.

Figure 3 has been updated to match the name between text and figure.

The first section of results is named “Environmental conditions and particle abundance” however there are no words on particle abundance. It is unclear why.

We apologize. It was included before because LISST and UVP data were included in a previous version of this ms. The section title has been corrected.

We now write in lines 377-385:

“Particle profiles by size class can be obtained, for instance, using laser in-situ scattering and transmissometry (LISST) and underwater vision profiler (UVP) 63-65. While we deployed both instruments during our cruise and obtained UVP particle profiles successfully (data not shown), the size of the particles with active N₂-fixing NCDs was 29 to 128 μm (Table S2), which falls below the detection range of the UVP and thus did not allow a particle profile extrapolation of particle-associated N₂ fixation rates for this study. Active N₂-fixing NCDs have been observed in larger particles (e.g., >210 μm, Harding et al. 2022), suggesting that a combination of LISST and UVP profiles is needed for extrapolation purposes in future studies as the size of particles bearing active NCDs may change among regions, seasons and depths.”

In the abstract, the long sentence in lines 30-35 should be split or re-worded for clarity.

The abstract has been rewritten for clarity. See below:

“Dinitrogen (N₂) fixation supports marine life through the supply of reactive nitrogen. Recent studies suggest that particle-associated non-cyanobacterial diazotrophs (NCDs) could contribute significantly to N₂ fixation contrary to the paradigm of diazotrophy as primarily driven by cyanobacterial genera. We examine the community composition of NCDs associated with suspended, slow, and fast-sinking particles in the North Pacific Subtropical Gyre. Suspended and slow-sinking particles showed a higher abundance of cyanobacterial diazotrophs than fast-sinking particles, while fast-sinking particles showed a higher diversity of NCDs including *Marinobacter*, *Oceanobacter* and *Pseudomonas*. Using single-cell mass spectrometry we find that Gammaproteobacteria N₂ fixation rates were higher on suspended and slow-sinking particles (up to 67 ± 48.54 fmol N cell⁻¹ d⁻¹), while putative NCDs’ rates were highest on fast-sinking particles (121 ± 22.02 fmol N cell⁻¹ d⁻¹). These rates are comparable to previous diazotrophic cyanobacteria

observations, suggesting that particle-associated NCDs may be important contributors to pelagic N₂ fixation.”

Ln 76: accounting for 50% of what? This sentence is not clear. I assume it is 50% of the signal for all diazotrophs however this need to be clarified.

Indeed, it has been clarified in lines 75-76:

“...with approximately 50% of *nifH* reads in the 0.5-5 μm size-fraction and up to 25% of total *nifH* reads in the 180 to 2,000 μm size fractions ^{32,33} “

Ln 78: should be “are often present”

Corrected.

Tom O. Delmont